

# A quantitative look on northwestern Tethyan foraminiferal assemblages, Campanian Nierental Formation, Austria

Erik Wolfgring[1,2] and Michael Wagreich[1]

[1] Department of Geodynamics and Sedimentology, University of Vienna, Vienna, Austria
[2] Department of Palaeontology, University of Vienna, Vienna, Austria

## ABSTRACT

Deposits spanning the *Radotruncana calcarata* Taxon Range Zone at the Postalm section, Northern Calcareous Alps (Austria) are examined quantitatively for foraminiferal assemblages, especially the planktonic group. This study focuses on establishing a high resolution record spanning an 800 ka long stratigraphic interval from the active continental margin of the Penninic Ocean. The Postalm section displays reddish limestone- marl alternations representing precession cycles. For this study, 26 samples were taken bed by bed to allow a "per-precession-cycle" resolution (i.e., a minimum sample distance of ∼20 ka). Samples from limestones as well as from marls were examined for foraminiferal assemblages. Data suggest a typical, open marine Campanian foraminiferal community. The >63 μm fraction is dominated by opportunist taxa, i.e., members of *Muricohedbergella* and biserial planktic foraminifera. *Archaeoglobigerina* and "*Globigerinelloides*" appear frequently and benthic foraminifera are very sparsely found. The share of globotruncanids, representing more complex morphotypes amongst planktonic foraminifera, is recorded with 5–10%. The state of preservation of foraminifera from the Postalm section is moderate to poor. Differences between samples from marls and samples from limestone are evident, but do not reveal evidence that there was an influence on the postdepositional microfossil communities. However, data from microfossils showing moderate to bad preservation can still offer valuable insight into the palaeoenvironment and biostratigraphy. Information gathered on the composition of the planktonic foraminiferal assemblage confirms a low-to-mid-latitude setting for the Postalm section. As well resolved records of Late Cretaceous foraminifera assemblages are rare, the examination of the *Radotruncana calcarata* Taxon Range Zone provides some insights into variations and short term changes during the very short period of 800 ka.

Corresponding author
Erik Wolfgring,
erik.wolfgring@univie.ac.at

## INTRODUCTION

The Late Cretaceous is a period recording major changes in the Earth's climate system. Trends in climate evolution reflect the transition from a mid-Cretaceous hothouse to a more moderate greenhouse during the later part of the Late Cretaceous (e.g., *Barrera & Savin, 1999; Huber, Norris & MacLeod, 2002; Friedrich et al., 2009; Friedrich, Norris & Erbacher, 2012; Hay & Floegel, 2012; Jung et al., 2013; Price et al., 2013; Linnert et al., 2014;*

*Sames et al., 2015*). Fundamental palaeoenvironmental processes influenced by climate and palaeoceanographic changes (e.g., the late Campanian—Maastrichtian cooling or sealevel changes of different magnitudes that affect chemical parameters like oxygen availability) can also be recognised as drivers behind modifications in the composition of foraminiferal assemblages, and especially planktonic foraminiferal communities, as discussed in this paper (*Premoli Silva & Sliter, 1999*; *Abramovich et al., 2003*; *Friedrich, Herrle & Hemleben, 2005*; *Falzoni et al., 2013*).

The mid-to-late Campanian—from the base of the *Contusotruncana plummerae* Zone at 79.2 Ma to the Campanian Maastrichtian boundary at 72.1 Ma (for the chronos-tratighraphic framework see *Huber, MacLeod & Tur, 2008*; *Anthonissen & Ogg, 2012*)—is generally considered a time interval with a highly diversified planktonic foraminifera fauna (*Premoli Silva & Sliter, 1999*; *Abramovich et al., 2003*). Prolonged evolution and development of foraminiferal communities is known from the middle Campanian to Maastrichtian, coinciding with the onset of the general end-Cretaceous cooling trend (*Hart, 1999*; *Premoli Silva & Sliter, 1999*; *Georgescu, 2005*). The radiation of Archaeoglo-bigerinidae and Rugoglobigerinidae, the diversification of biserial planktonic taxa, the appearance of complex morphotypes in globotruncanids—all of these are developments during the mid to late Campanian to Maastrichtian (*Hart, 1999*; *Premoli Silva & Sliter, 1999*; *Georgescu, 2005*).

Few high-resolution studies on general evolutionary trends, visible in the quantitative data from Campanian foraminifera communities exist. In general, most of quantitative studies on Late Cretaceous foraminiferal assemblages focus on developments around stage boundaries and/or events (e.g., *Huber et al., 1999*; *Arz & Molina, 2001*; *Odin & Lamaurelle, 2001*; *Petrizzo, 2002a*; *Caron et al., 2006*; *Elamri & Zaghbib-Turki, 2014*; *Elamri, Farouk & Zaghbib-Turki, 2014*; *Reolid et al., 2015*). The vast majority deals with the Cretaceous- Paleogene turnover (e.g., *Abramovich, Almogi-Labin & Benjamini, 1998*; *Li & Keller, 1998*; *Arenillas et al., 2000*; *Abramovich & Keller, 2002*; *Karoui-Yaakoub, Zaghbib-Turki & Keller, 2002*; *Premoli Silva, Emeis & Robertson, 2005*; *Gallala et al., 2009*; *Beiranvand & Ghasemi-Nejad, 2013*, see also *Pardo & Keller, 2008* for a compilation of selected quantitative databases on the Cretaceous-Paaeogene boundary).

An almost complete Santonian-lower Maastrichtian succession is recorded in pelagic to hemipelagic deposits at the Postalm section, Austria, at the NW margin of the Tethys (Fig. 1). The study of *Wagreich, Hohenegger & Neuhuber (2012)* addresses the bios-tratigraphy, as well as the astronomical calibration of the *R. calcarata* Zone in the mid-Campanian at Postalm. With its rather short duration of only 800 ka (806.3 ka in the study of *Wagreich, Hohenegger & Neuhuber, 2012*; *Robaszynski & Mzoughi, 2010*, give a mean duration of 790 ka), and the distinct morphology of the nominative taxon, the *Radotruncana calcarata* Taxon-range Zone is considered a well-established, easily recognisable and reliable time interval in Late Cretaceous chronostratigraphy of the Tethyan Realm (e.g., *Robaszynski et al., 1984*; *Chungkham & Jafar, 1998*; *Premoli Silva, Spezzaferi & D'Angelantonio, 1998*; *Puckett & Mancini, 1998*; *Huber, MacLeod & Tur, 2008*; *Wendler et al., 2011*).
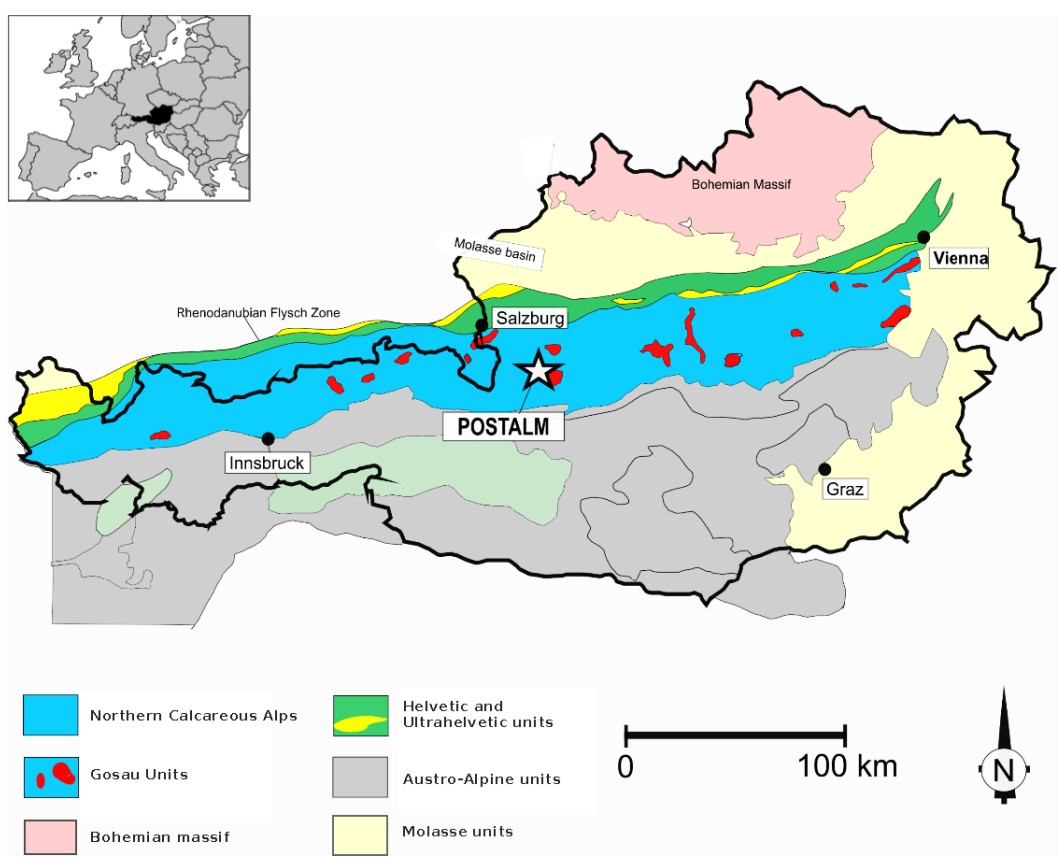

**Figure 1** **Geological sketch-map of the Austrian Alps.** The Postalm section is situated some kilometres south of the city of Salzburg. The inset explains the geographical context.

*Wolfgring, Hohenegger & Wagreich (in press)* reported biostratigraphic and qualitative foraminiferal data based on presence–absence data from two sections on opposite margins of the Penninic Ocean, including data from the Postalm section. The latter displayed a remarkably static composition of foraminiferal assemblages in the *Radotruncana calcarata* Zone. The sudden appearance and disappearance of the zonal marker and the disappearance of *Globotruncanita elevata* were the only biostratigraphic events recorded in the planktonic foraminifera record from this section.

In this work we present a quantitative study on planktonic foraminiferal assemblages in the *R. calcarata* Zone at Postalm section. With 26 samples in the 803.6 ka long interval, this high-resolution study gives information on the composition of typical Tethyan pelagic assemblages. This work deals with subtle changes in north-western planktonic foraminiferal communities on the brink of the Late Cretaceous cooling and major faunal turnover-events (*Premoli Silva & Sliter, 1999*). With adding the quantitative aspect of a comparatively short episode of the Tethyan Campanian to the presence–absence data assessed in *Wolfgring, Hohenegger & Wagreich (in press)* we aim at a better understanding of small scale changes in the foraminiferal record (e.g., the extinction of *R. calcarata* and *Gta. elevata*).

## Geological setting

Units forming the Northern Calcareous Alps (NCA) were deposited along the northern margin of the Austroalpine domain on the Adriatic microplate (*Wagreich, 1993*), at the southern margin of the Penninic Ocean ("Alpine Tethys" of *Stampfli & Borel, 2002*; *Handy et al., 2010*), which was a north-western part of the Tethys oceanic system (see also *Neuhuber et al., 2007*).

Within the system of the NCA, the Upper Cretaceous to Paleogene Gosau Group is characterised by the terrestrial to shallow marine Lower Gosau Subgroup and the deepwater deposits of the Upper Gosau Subgroup. The Lower Gosau Subgroup of Turonian to Santonian age filled pull-apart basins alongside an oblique subduction—strike-slip zone (*Wagreich & Decker, 2001*). After a short phase of tectonically induced uplift of the NCA, rapid subsidence processes resulted in the sedimentation of the pelagic, hemipelagic and turbiditic Upper Gosau Subgroup, comprising strata of Santonian/Campanian to Eocene age (*Wagreich, 1993*; *Krenmayr, 1999*; *Wagreich et al., 2011*; *Hofer et al., 2011*).

The Postalm section (coordinates WGS 84 013°23′11″E; 47°36′44″N) belongs to the Nierental Formation of the Upper Gosau Subgroup (Fig. 1) (*Krenmayr, 1996*; *Wagreich & Krenmayr, 2005*; *Wagreich, Hohenegger & Neuhuber, 2012*). The Nierental Formation was originally deposited at palaeolatitudes of approximately 35—30°N, alongside the southern margin of the Penninic Ocean (Fig. 2). The Santonian to Maastrichtian succession at Postalm is characterised by distinct marly limestone—marl cycles and records an upper to middle bathyal depositional environment (*Wagreich, Hohenegger & Neuhuber, 2012*) (Fig. 3). Marly limestones can be classified as foraminiferal packstone. The Postalm section is interpreted as a pelagic to hemipelagic depositional environment well above the CCD. The section was part of a northward deepening slope within the NCA with bathyal water-depths (*Wagreich & Krenmayr, 2005*; *Wolfgring, Hohenegger & Wagreich, in press*).

The deposits recorded at the Postalm section are interpreted as Cretaceous Oceanic Red Beds (CORB), indicating overall well oxygenated bottom waters (*Hu et al., 2005*; *Wagreich & Krenmayr, 2005*). The sediment accumulation rate is estimated to be 20 mm/ka (*Wagreich, Hohenegger & Neuhuber, 2012*).

For more detailed information on the geological setting at Postalm section, the reader is referred to *Wagreich, Hohenegger & Neuhuber (2012)*.

# MATERIAL AND METHODS

## Sampling and samples preparation

The *Radotruncana calcarata* Taxon Range Zone (TRZ) was sampled bed-by-bed, following biostratigraphic investigation of *Wagreich, Hohenegger & Neuhuber (2012)* and *Neuhuber et al. (2015)*. No standard sampling distance was applied. Figure 4 gives an overview on the stratigraphic framework and the location of sample-spots used in *Wolfgring, Hohenegger & Wagreich (in press)* and in the present study. Twentysix samples from marls and marly limestones were processed to obtain quantitative data. Marl and marly-limestone samples were soaked in the tenside Rewoquad© for 24 h and then thoroughly rinsed with water. Samples were thereafter soaked in hydrogen peroxide (35%) for 1 h

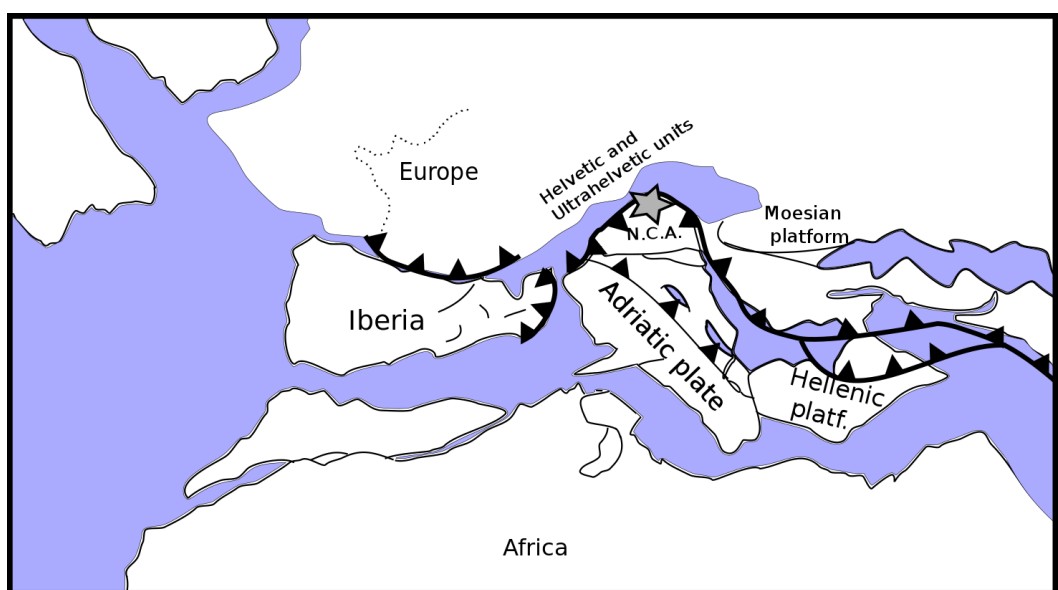

**Figure 2** **Palaeogeographic reconstruction of the Penninic realm (redrawn, simplified and modified from *Schettino & Turco, 2011*).** The Postalm section (grey star) is located in the Northern Calcareous Alps (N.C.A.) on the southern active margin of the Penninic Ocean.

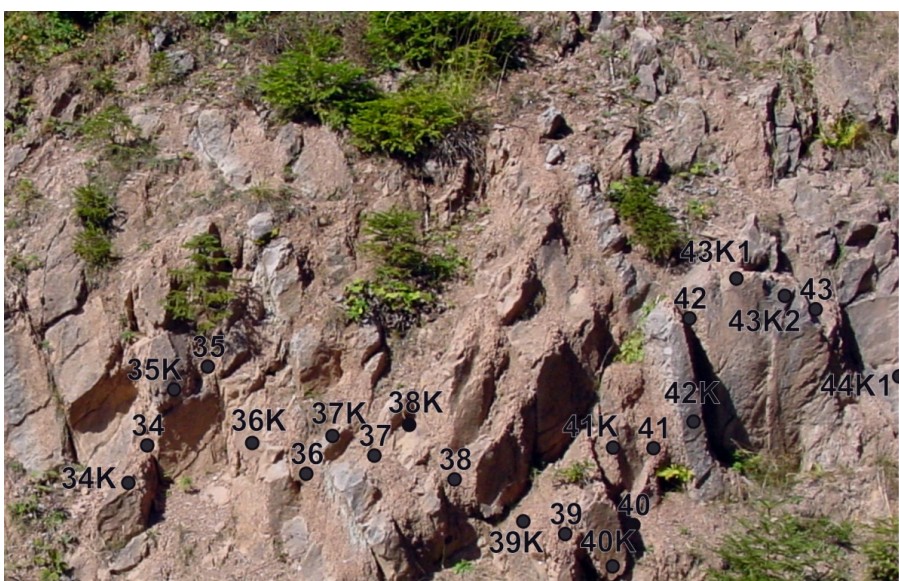

**Figure 3** **Detail of cyclic marl-marly limestone alternations at Postalm depicting the older part of the *R. calcarata* interval.** Black dots show sample locations.

and wet sieved. Firm foraminifera packstone required intense treatment; cooking the samples in hydrogen peroxide for ten minutes and the repeated use of tensides was mandatory. Disaggregated samples were washed over 63 µm and 125 µm mesh sieves. The residues were dried overnight at 60°.

Quantitative data were assessed using the >63 µm size fractions. "Larger" foraminifera (>125 µm) were assigned genus and species, while the 63–125 µm fraction is mostly

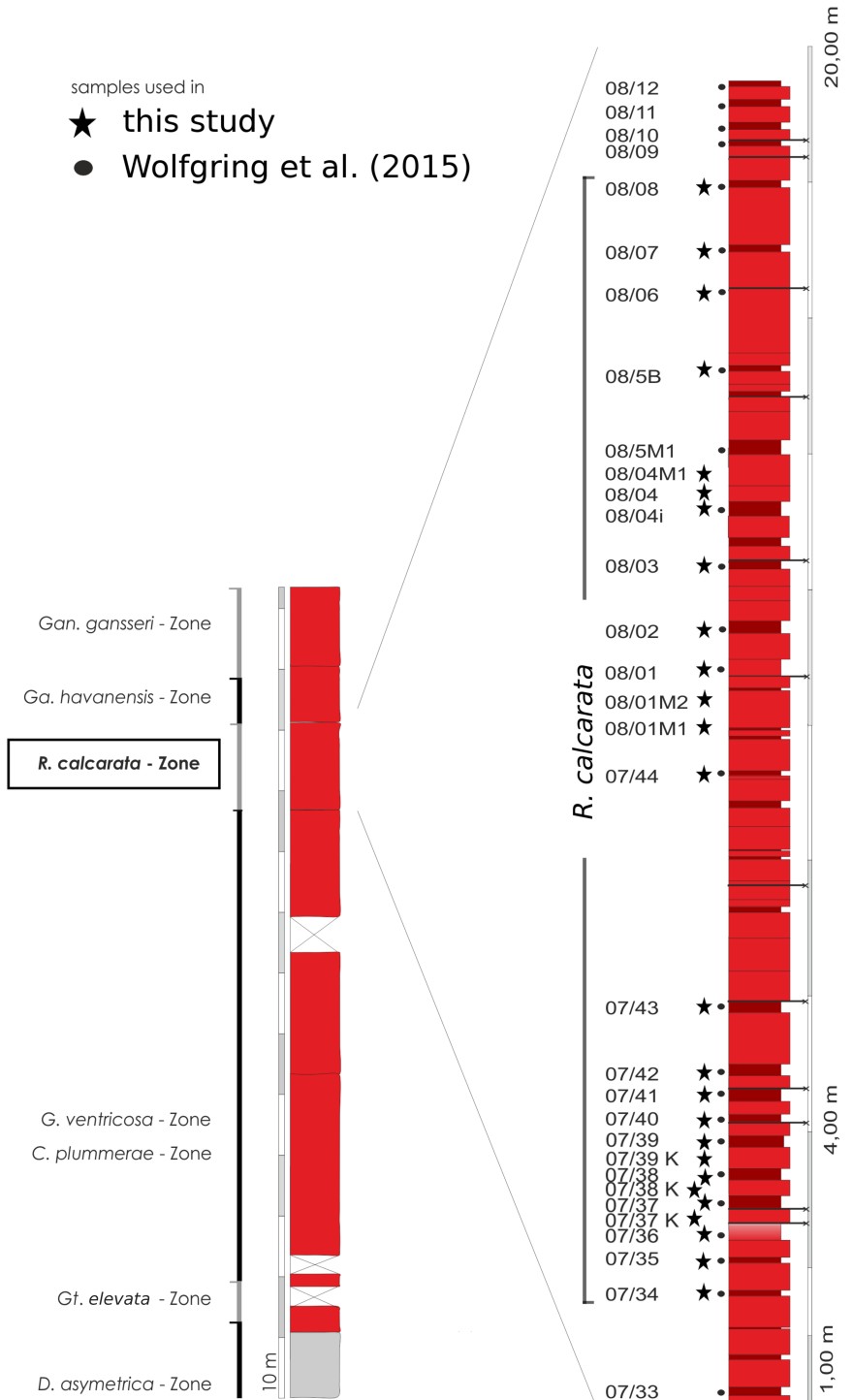

**Figure 4** **Overview of the geological setting at Postalm section and the *R. calcarata* interval in detail.**
Sample points from this the present study are flagged with stars. Samples used for the assessment of
presence-absence data in *Wolfgring, Hohenegger & Wagreich (in press)* are flagged with black dots.

discussed on genus level, as, in some cases, the state of preservation did not permit the identification of taxonomically relevant features. According to micropalaeontological standard procedures, per sample at least some 300 specimen of both planktic and benthic foraminifera were counted. Data were obtained from marls, as well as marly limestones.

The tool PanPlot 2 (*Sieger & Grobe, 2013*) was used to visualise foraminifera abundances in Postalm section.

Samples and microslides are stored in the Earth Science collections at the Department of Geodynamics and Sedimentology, University of Vienna.

## Palaeodepth estimates

Depositional palaeo-waterdepth was calculated applying the method published in *Van der Zwaan, Jorissen & De Stigter (1990)*. Here, the palaeodepth in meters is estimated as:

$$D(m) = e^{3.58718 + (0.03534 \times P)} \tag{1}$$

where $D(m)$ is the estimated palaeodepth in metres, $e$ is the mathematical constant Euler's number and $P$ the ratio of planktonic/benthic foraminifera. The calculation of $P$ excludes benthic foraminiferal taxa from the analysis that are not directly dependent on the flux of organic matter to the seafloor (*Van der Zwaan, Jorissen & De Stigter, 1990*; *Kopecká, 2012*). This is the case with a mono-specific mass occurrence of *Nothia* sp. at the Postalm section. As episodic blooms in this taxon could be a possible reason for the high share in some samples this taxon has been excluded from this analysis.

In addition to the method of *Van der Zwaan, Jorissen & De Stigter (1990)* that relies on the quantitative data assessed in this study, presence–absence data of benthic foraminifera recorded at the Postalm section (*Wolfgring, Hohenegger & Wagreich, in press*) was used to calculate palaeo water depth applying the method of *Hohenegger (2005)*. The basic formula for estimating the depth gradient is given as:

$$g = \sum_{i=1}^{m} l_i d_i^{-1} \left/ \sum_{i=1}^{m} d_i^{-1} \right. \tag{2}$$

where $i_i$ is the mean depth and $d_i$ the distribution range along the taxon's depth range and $g$ the estimated palaeodepth.

Depth ranges of benthic foraminifera (see Appendix S1) are inferred from the palaeoslope model of *Nyong & Olssen (1984)* with depth ranges for Campanian-Maastrichtian benthic foraminifera along the Atlantic coast, as well as the bathymetric ranges of benthic foraminifera of *Sliter & Baker (1972)*, *Speijer & Van der Zwaan (1996)*, *Kaminski & Gradstein (2005)*, *Valchev (2006)* as well as *Holbourn, Henderson & MacLeod (2013)*.

## Taxonomic remarks/methods and the preservation of microfossils

With few exceptions, the state of preservation in the investigated samples can be considered moderate to poor. Most spiral and trochospiral planktic and benthic forms appear with fully intact tests. No signs of dissolution were recorded. Some planktonic foraminifera show evidence of recrystallisation and carbonate infilling. Elongated forms frequently appear fragmented. However, the state of preservation did not allow the

definite taxonomic assignment of some individuals to species level. Thus, morphogroups for certain taxa were established.

Some double keeled, biconvex globotruncanid taxa (*Globotruncana arca, G. lapparenti, G. orientalis*) have subsequently been merged to *Globotruncana arca-lapparenti-orientalis,* as morphological transitions were observed. Some biserial planktonic specimens displayed a very bad state of preservation, rendering the identification of some individuals on species level impossible. These specimens were aggregated into the group *Planoheterohelix* spp. Biserial planktonic taxa with reniform chambers were pooled under *Laeviheterohelix* spp.

Planktonic foraminiferal taxonomy predominantly follows *Nederbragt (1991)*, *Robaszynski & Caron (1995)* and *Premoli Silva & Verga (2004)*. *Georgescu & Huber (2009)* and *Petrizzo, Falzoni & Premoli Silva (2011)*, Genera with their taxonomy under revision appear in quotes. Some significant taxa of the section are pictured in *Wolfgring, Hohenegger & Wagreich (in press)*.

To define the trophic characteristics of the investigated area, we determined the distribution of $r$- and $K$-strategists. The $r$-strategists are generally considered to be opportunists and adapted to eutrophic or unstable conditions; $K$-strategists represent more complex morphotypes that favour stable, rather oligotrophic environments (*Premoli Silva & Sliter, 1999*; *Petrizzo, 2002a*; *Petrizzo, 2002b*; *Gebhardt et al., 2010*).

Benthic foraminiferal taxa were not assigned genus and species. Benthic foraminifera are extremely sparse in standard quantitative data. The number of benthic specimen per sample was recorded but we refrained from any taxonomic assignment. Presence–absence data on the benthic foraminiferal record (from *Wolfgring, Hohenegger & Wagreich, in press*) is available in the Appendix S1.

## RESULTS

### Quantitative data

The quantitative investigation of the *R. calcarata TRZ* displays the composition of a typical mid Campanian low to mid-latitude Tethyan foraminiferal community and records at least 42 different planktonic foraminiferal species in 15 genera (Fig. 5). Benthic foraminifera are very sparse in quantitative data. Table 1 displays the number of specimens as well as the relative abundances of foraminiferal species. The foraminiferal assemblage at Postalm is dominated by members of heterohelicids and the genus *Muricohedbergella*. These two groups account for up to 80% of the total assemblage. Small heterohelicid taxa are dominated by *Planoheterohelix globulosa*. We can confirm the presence of *Pseudotextularia nuttalli, Gublerina rajagopalani, Ph. striata, Spiroplecta navarroensis* and *Guembelitria* sp. However, the groups *Planoheterohelix* spp. and *Laeviheterohelix* spp. comprise a relatively high proportion of the assemblage. Multiserial or flaring heterohelicids were not detected.

The genus *Muricohedbergella* is represented by the species *M. holmdelensis* and *M. monmouthensis* in varying numbers. The share of this group slightly increases towards the top of the *R. calcarata* TRZ.

Globotruncanids are less abundant and represent between 1 and 12 percent. This group is represented by the genera *Globotruncana, Globotruncanella, Globotruncanita,*
**Table 1** Displays the proportional frequencies of foraminiferal taxa per sample at the Postalm section.

| Species | 08/08 | 08/07 | 08/06 | 08/5b | 08/04 | 08/4i | 08/4M1 | 08/03 | 08/02 | 08/01 | 08/1M2 | 08/1M1 | 07/44 | 07/43 | 07/42K | 07/41 | 07/40 | 07/39 | 07/39K | 07/38 | 07/38K | 07/37 | 07/37K | 07/36 | 07/35 | 07/34 |
|---|---|---|---|---|---|---|---|---|---|---|---|---|---|---|---|---|---|---|---|---|---|---|---|---|---|---|
| *Archaeoglobigerina blowi* | – | – | – | – | – | – | – | – | – | – | – | – | – | – | – | – | – | – | – | – | – | X | – | X | – | X |
|  | – | – | – | – | – | – | – | – | – | – | – | – | – | – | – | – | – | – | – | – | – | – | – | – | – | – |
| *A. cretacea* | – | – | – | – | 20 | – | 14 | – | 7 | – | 2 | 28 | – | – | 2 | – | – | – | – | – | – | – | – | – | – | 1 |
|  | – | – | – | – | (4%) | – | (4%) | – | (2%) | – | (<1%) | (7%) | – | – | (<1%) | – | – | – | – | – | – | – | – | – | – | (<1%) |
| *Archaeoglobigerina* spp. | 48 | 8 | 7 | 2 | 1 | 1 | 4 | 1 | 11 | 16 | 22 | 1 | – | – | 36 | 9 | 3 | 6 | 11 | 15 | 5 | 12 | 3 | 15 | 16 | 10 |
|  | (6%) | (3%) | (4%) | (1%) | (<1%) | (<1%) | (1%) | (<1%) | (3%) | (5%) | (5%) | (<1%) | – | – | (8%) | (3%) | (1%) | (1%) | (4%) | (6%) | (1%) | (4%) | (1%) | (4%) | (2%) | (3%) |
| *Contusotruncana fornicata* | 14 | 1 | 1 | 1 | 1 | 1 | 1 | 4 | 1 | 1 | 1 | 1 | 5 | 1 | 1 | 2 | 1 | 3 | 1 | 2 | 5 | 1 | 2 | 2 | 1 | 1 |
|  | (2%) | (<1%) | (1%) | (<1%) | (<1%) | (<1%) | (<1%) | (1%) | (<1%) | (<1%) | (<1%) | (<1%) | (2%) | (<1%) | (<1%) | (1%) | (<1%) | (<1%) | (<1%) | (1%) | (1%) | (<1%) | (1%) | (1%) | (<1%) | (<1%) |
| *C. morozovae* | – | – | – | – | – | – | – | – | 0 | – | – | – | – | – | – | – | – | – | – | 1 | – | – | – | – | – | – |
|  | – | – | – | – | – | – | – | – | (0%) | – | – | – | – | – | – | – | – | – | – | (0%) | – | – | – | – | – | – |
| *C. patelliformis* | 10 | 7 | 1 | 1 | 1 | – | 1 | 4 | 1 | 1 | 1 | 1 | 2 | 1 | – | 1 | 1 | 1 | 1 | 3 | – | 8 | 3 | 1 | 3 | 1 |
|  | (1%) | (2%) | (1%) | (<1%) | (<1%) | – | (<1%) | (1%) | (<1%) | (<1%) | (<1%) | (<1%) | (1%) | (<1%) | – | (<1%) | (<1%) | (<1%) | (<1%) | (1%) | – | (3%) | (1%) | (<1%) | (<1%) | (<1%) |
| *C. plummerae* | X | X | – | X | – | – | – | – | X | – | – | – | – | X | – | – | – | – | – | X | – | – | – | – | – | – |
|  | – | – | – | – | – | – | – | – | – | – | – | – | – | – | – | – | – | – | – | – | – | – | – | – | – | – |
| *Contusotruncana* sp. | – | – | 1 | 1 | – | 1 | – | 1 | – | 1 | – | – | – | – | – | – | – | – | – | – | – | – | – | – | – | – |
|  | – | – | (1%) | (<1%) | – | (<1%) | – | (<1%) | – | (<1%) | – | – | – | – | – | – | – | – | – | – | – | – | – | – | – | – |
| *Globotruncana arca–lapparenti–orientalis* | 37 | 13 | 2 | 2 | 3 | 4 | 3 | 4 | 1 | 1 | 3 | 1 | 6 | 1 | 3 | 1 | 2 | 13 | 1 | 1 | 4 | 3 | 4 | 3 | 14 | 2 |
|  | (5%) | (4%) | (1%) | (1%) | (1%) | (2%) | (1%) | (1%) | (<1%) | (<1%) | (1%) | (<1%) | (2%) | (<1%) | (1%) | (<1%) | (1%) | (2%) | (<1%) | (<1%) | (1%) | (1%) | (1%) | (1%) | (2%) | (1%) |
| *G. aff. conica* | 3 | – | – | – | – | – | – | – | – | – | – | – | – | – | – | – | – | – | – | – | – | – | – | – | – | – |
|  | (<1%) | – | – | – | – | – | – | – | – | – | – | – | – | – | – | – | – | – | – | – | – | – | – | – | – | – |
| *G. atlantica* | – | – | – | – | – | – | – | – | – | – | – | – | – | – | – | – | – | – | – | – | – | – | – | – | – | – |
|  | – | – | – | – | – | – | – | – | – | – | – | – | – | – | – | – | – | – | – | – | – | – | – | – | – | – |
| *G. bulloides* | – | 6 | 1 | – | – | – | 1 | – | – | – | – | 1 | – | – | – | – | – | – | – | – | – | – | – | – | – | 1 |
|  | – | (2%) | (1%) | – | – | – | (<1%) | – | – | – | – | (<1%) | – | – | – | – | – | – | – | – | – | – | – | – | – | (<1%) |
| *G. falsostuarti* | – | – | 1 | 1 | – | – | – | 1 | – | – | 1 | 1 | – | – | – | 1 | 1 | – | – | – | – | 1 | – | 1 | 3 | 1 |
|  | – | – | (<1%) | (<1%) | – | – | – | (<1%) | – | – | (<1%) | (<1%) | – | – | – | (<1%) | (<1%) | – | – | – | – | (<1%) | – | (<1%) | (<1%) | (<1%) |
| *G. linneiana* | 19 | 1 | 1 | 1 | 2 | 1 | 1 | 2 | 2 | 1 | 1 | 1 | 2 | 1 | – | – | 1 | 2 | 1 | – | – | 1 | 1 | 1 | 1 | 1 |
|  | (2%) | (<1%) | (<1%) | (<1%) | (<1%) | (<1%) | (<1%) | (1%) | (1%) | (<1%) | (<1%) | (<1%) | (1%) | (1%) | – | – | (<1%) | (<1%) | (<1%) | – | – | (<1%) | (<1%) | (<1%) | (<1%) | (<1%) |
| *G. stuartiformis* | – | 1 | – | 1 | – | – | – | 1 | – | 1 | – | 1 | – | – | 1 | – | – | – | – | 1 | – | – | 1 | – | – | 2 |
|  | – | (0%) | – | (<1%) | – | – | – | (<1%) | – | (0%) | – | (<1%) | – | – | (<1%) | – | – | – | – | (<1%) | – | – | (<1%) | – | – | (1%) |
| *G. tricarinata* | 5 | – | – | – | 1 | 1 | – | 1 | – | – | – | – | – | – | – | – | – | – | – | – | – | – | – | – | – | 1 |
|  | (1%) | – | – | – | (<1%) | (<1%) | – | (<1%) | – | – | – | – | – | – | – | – | – | – | – | – | – | – | – | – | – | (0%) |

| Species | 08/08 | 08/07 | 08/06 | 08/5b | 08/04 | 08/4i | 08/4M1 | 08/03 | 08/02 | 08/01 | 08/1M2 | 08/1M1 | 07/44 | 07/43 | 07/42K | 07/41 | 07/40 | 07/39 | 07/39K | 07/38 | 07/38K | 07/37 | 07/37K | 07/36 | 07/35 | 07/34 |
|---|---|---|---|---|---|---|---|---|---|---|---|---|---|---|---|---|---|---|---|---|---|---|---|---|---|---|
| G. ventricosa | – | – | 1 | 1 | 1 | 1 | – | 1 | 2 | 1 | 1 | 1 | 2 | 1 | 2 | 1 | – | – | 1 | – | – | – | – | 2 | 7 | 1 |
|  | – | – | (1%) | (<1%) | (<1%) | (<1%) | – | (<1%) | (1%) | (<1%) | (<1%) | (<1%) | (1%) | (<1%) | (<1%) | (<1%) | – | – | (<1%) | – | – | – | – | (<1%) | (1%) | (<1%) |
| G. mariei | – | – | – | – | – | – | – | – | – | – | – | – | – | – | – | – | – | – | – | – | – | – | – | 1 | – | – |
|  | – | – | – | – | – | – | – | – | – | – | – | – | – | – | – | – | – | – | – | – | – | – | – | (0%) | – | – |
| Globotruncana sp. | – | 4 | 1 | 1 | 2 | – | 2 | – | 3 | 1 | 3 | 1 | – | – | 2 | – | – | – | – | – | – | – | – | 2 | 4 | 2 |
|  | – | (1%) | (<1%) | (<1%) | (<1%) | – | (1%) | – | (1%) | (<1%) | (1%) | (<1%) | – | – | (1%) | – | – | – | – | – | – | – | – | (0%) | (1%) | (1%) |
| Globotruncanella havanensis | – | – | – | – | – | 1 | – | – | – | – | – | – | – | – | – | – | – | 1 | – | – | – | – | – | – | – | – |
|  | – | – | – | – | – | (<1%) | – | – | – | – | – | – | – | – | – | – | – | (<1%) | – | – | – | – | – | – | – | – |
| Ga. pschadae/sp | – | – | – | 1 | – | – | – | – | – | – | – | – | – | – | – | – | – | – | – | – | – | – | – | – | – | – |
|  | – | – | – | (<1%) | – | – | – | – | – | – | – | – | – | – | – | – | – | – | – | – | – | – | – | – | – | – |
| Globotruncanita sp. | 1 | – | – | – | – | – | – | – | – | – | – | – | – | – | – | – | 1 | 1 | – | 1 | – | – | – | – | 1 | – |
|  | (<1%) | – | – | – | – | – | – | – | – | – | – | – | – | – | – | – | (<1%) | (<1%) | – | (<1%) | – | – | – | – | (<1%) | – |
| Gta. elevata | – | – | – | – | – | – | – | – | – | – | – | – | – | – | – | – | X | – | X | – | – | – | – | X | – | X |
|  | – | – | – | – | – | – | – | – | – | – | – | – | – | – | – | – | – | – | – | – | – | – | – | – | – | – |
| Gta. subspinosa | – | – | – | – | – | – | – | – | – | – | – | – | – | – | – | X | – | – | – | – | – | – | – | – | – | – |
|  | – | – | – | – | – | – | – | – | – | – | – | – | – | – | – | – | – | – | – | – | – | – | – | – | – | – |
| Guembelitra sp. | – | – | 6 | – | – | – | – | – | – | – | – | – | – | – | – | – | – | 10 | 1 | – | 12 | – | – | 2 | – | – |
|  | – | – | (3%) | – | – | – | – | – | – | – | – | – | – | – | – | – | – | (2%) | (<1%) | – | (3%) | – | – | (<1%) | – | – |
| Muricohedbergella holmdelensis | 54 | 22 | 13 | 20 | 38 | 17 | 27 | 28 | 16 | 28 | 44 | 31 | 39 | 51 | 45 | 39 | 47 | 64 | 26 | 56 | 37 | 37 | 32 | 69 | 42 | 104 |
|  | (7%) | (7%) | (6%) | (6%) | (7%) | (7%) | (7%) | (8%) | (4%) | (9%) | (11%) | (8%) | (12%) | (21%) | (9%) | (12%) | (16%) | (10%) | (10%) | (21%) | (8%) | (14%) | (9%) | (19%) | (6%) | (34%) |
| M. monmouthensis | 267 | 101 | 57 | 99 | 200 | 79 | 102 | 106 | 104 | 105 | 130 | 142 | 148 | 137 | 125 | 117 | 161 | 280 | 91 | 55 | 45 | 86 | 114 | 69 | 206 | 10 |
|  | (35%) | (33%) | (29%) | (31%) | (39%) | (34%) | (28%) | (32%) | (26%) | (34%) | (32%) | (38%) | (47%) | (56%) | (27%) | (35%) | (57%) | (46%) | (34%) | (21%) | (10%) | (32%) | (32%) | (19%) | (30%) | (3%) |
| Planoheterohelix globulosa | 168 | 64 | 54 | 125 | 144 | 78 | 133 | 99 | 165 | – | 127 | 152 | 86 | 25 | 214 | 117 | 27 | 142 | 128 | 67 | 174 | 39 | 160 | 106 | 165 | 92 |
|  | (22%) | (21%) | (28%) | (40%) | (28%) | (34%) | (36%) | (30%) | (41%) | – | (32%) | (41%) | (27%) | (10%) | (45%) | (35%) | (9%) | (23%) | (47%) | (26%) | (40%) | (15%) | (45%) | (29%) | (24%) | (30%) |
| Ph. pupa | – | – | – | – | – | – | – | 1 | – | – | – | – | – | – | – | – | – | 1 | – | – | – | – | – | – | – | – |
|  | – | – | – | – | – | – | – | (<1%) | – | – | – | – | – | – | – | – | – | (<1%) | – | – | – | – | – | – | (<1%) | – |
| Gublerina rajagopalani | – | – | – | – | – | – | – | – | – | – | – | – | – | – | – | – | 1 | – | – | – | – | – | – | 1 | – | – |
|  | – | – | – | – | – | – | – | – | – | – | – | – | – | – | – | – | (<1%) | – | – | – | – | – | – | (<1%) | – | (<1%) |
| Ph. striata | 6 | 3 | 1 | 17 | – | 1 | – | 5 | 17 | 1 | – | 1 | – | 1 | – | – | 1 | 2 | 1 | 1 | 2 | 31 | 6 | – | 54 | 11 |
|  | (1%) | (1%) | (1%) | (5%) | – | (<1%) | – | (1%) | (4%) | (<1%) | – | (<1%) | – | (<1%) | – | – | (<1%) | (<1%) | (<1%) | (1%) | (1%) | (12%) | (2%) | – | (8%) | (4%) |
| Planoheterohelix spp. | 84 | 43 | 13 | 5 | 53 | 12 | 30 | 51 | 28 | 130 | 26 | 1 | 2 | 5 | 3 | 33 | 17 | 33 | 1 | 49 | 95 | 21 | 1 | 72 | 85 | 37 |
|  | (11%) | (14%) | (6%) | (2%) | (10%) | (5%) | (8%) | (15%) | (7%) | (42%) | (6%) | (<1%) | (1%) | (2%) | (1%) | (10%) | (6%) | (5%) | (<1%) | (18%) | (22%) | (8%) | (<1%) | (20%) | (12%) | (12%) |
| "Spiroplecta" navarroensis | – | – | – | – | – | 1 | – | – | – | – | – | – | – | – | – | – | 1 | – | – | – | – | – | – | 1 | 54 | – |
|  | – | – | – | – | – | (<1%) | – | – | – | – | – | – | – | – | – | – | (<1%) | – | – | – | – | – | – | (<1%) | (8%) | – |

**Table 1** (*continued*)

| Species | Sample | | | | | | | | | | | | | | | | | | | | | | | | | |
|---|---|---|---|---|---|---|---|---|---|---|---|---|---|---|---|---|---|---|---|---|---|---|---|---|---|---|
| | 08/08 | 08/07 | 08/06 | 08/5b | 08/04 | 08/4i | 08/4M1 | 08/03 | 08/02 | 08/01 | 08/1M2 | 08/1M1 | 07/44 | 07/43 | 07/42K | 07/41 | 07/40 | 07/39 | 07/39K | 07/38 | 07/38K | 07/37 | 07/37K | 07/36 | 07/35 | 07/34 |
| *Leaviheterohelix* spp. | 9 (1%) | 4 (1%) | 0 (<1%) | – | – | 12 (5%) | 1 (0%) | 2 (1%) | 9 (2%) | – | 15 (4%) | – | 10 (3%) | 5 (2%) | – | 6 (2%) | 6 (2%) | 8 (1%) | – | 8 (3%) | 2 (1%) | – | – | – | – | – |
| "*Globigerinelloides*" *bolli* | 18 (2%) | 15 (5%) | 13 (7%) | 27 (9%) | – | 1 (<1%) | – | 6 (2%) | – | – | 1 (<1%) | – | 1 (<1%) | – | – | – | 11 (4%) | 2 (<1%) | 2 (1%) | 1 (1%) | – | – | 2 (1%) | 6 (2%) | 6 (1%) | 7 (2%) |
| "*G*". *prairiehillensis* | 15 (2%) | 10 (3%) | 17 (9%) | 7 (2%) | 48 (9%) | 17 (7%) | 45 (12%) | 15 (4%) | 34 (8%) | 18 (6%) | 22 (5%) | 10 (3%) | 14 (4%) | 14 (6%) | 36 (8%) | 6 (2%) | 2 (1%) | 2 (<1%) | 2 (1%) | 1 (1%) | 37 (8%) | 27 (10%) | 26 (7%) | 6 (2%) | 32 (5%) | 21 (7%) |
| "*G*". *ultramicrus* | – | – | – | – | 1.00 (<1%) | – | – | – | – | – | – | – | – | – | – | – | – | – | – | – | – | – | – | – | – | – |
| "*Globigerinelloides*" sp. | 1 (<1%) | – | – | – | – | – | – | – | – | – | – | – | – | – | – | – | – | – | – | – | – | – | – | – | – | – |
| *Pseudotextularia nuttalli* | – | – | 1 (1%) | – | – | – | – | 1 (<1%) | – | – | – | – | 1 (<1%) | – | – | – | – | – | – | – | – | – | – | 1 (<1%) | – | 1 (<1%) |
| *Radotruncana calcarata* | 1 (<1%) | X – | X – | X – | X – | 1 (<1%) | X – | X – | X – | 1 (<1%) | X – | X – | X – | X – | X – | X – | 1 (<1%) | 1 (<1%) | X – | X – | 1 (<1%) | X – | X – | X – | X – | X – |
| *Rugoglobigerina* sp. | – | – | – | – | – | – | – | – | – | – | 2 (1%) | – | – | – | – | – | 1 (0%) | – | – | – | – | – | – | – | 1 (<1%) | – |
| *Schackoina* sp. | – | 2 (1%) | – | 2 (1%) | – | – | – | 2 (1%) | – | 2 (1%) | – | – | – | – | – | – | – | – | – | – | – | – | – | – | – | – |
| Benthic foraminifera | 5 (1%) | 1 (<1%) | 1 (1%) | 1 (<1%) | 1 (<1%) | 2 (1%) | 1 (<1%) | 1 (<1%) | 1 (<1%) | 1 (<1%) | 1 (<1%) | 1 (<1%) | – | – | 1 (<1%) | 1 (<1%) | 1 (<1%) | 39 (6%) | 1 (<1%) | 1 (<1%) | 17 (4%) | 2 (1%) | – | 2 (<1%) | 3 (<1%) | – |
| **Total** | 765 | 308 | 194 | 315 | 518 | 232 | 368 | 337 | 405 | 311 | 403 | 376 | 319 | 246 | 474 | 335 | 285 | 611 | 271 | 264 | 439 | 268 | 357 | 361 | 699 | 309 |

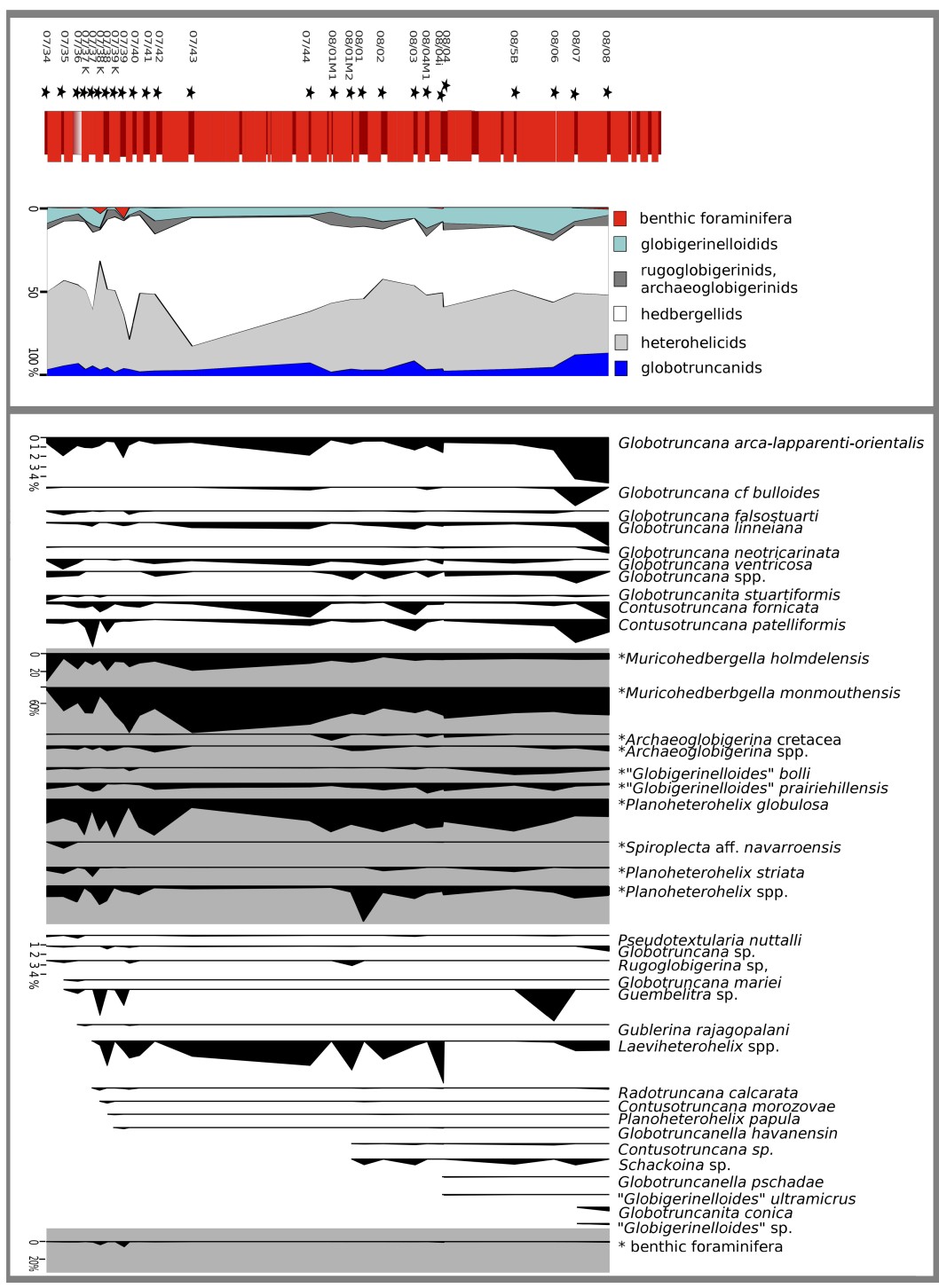

**Figure 5** **Relative abundance of foraminiferal species in the *R. calcarata* interval at Postalm section (>63 μm).** Species marked with an asterisk are necessarily displayed at a different scale. Species are in order of their stratigraphical appearance.

**Table 2** Life strategies of planktonic foraminifera at Postalm section (after *Premoli Silva & Sliter, 1999*; *Petrizzo, 2002a*; *Petrizzo, 2002b*).

| |
|---|
| *K*-selected: |
| *Contusotruncana* aff. *morozovae* |
| *C. patelliformis* |
| *C. plummerae* |
| *Contusotruncana fornicata* |
| *Contusotruncana* sp. |
| *Globotruncana arca-lapparenti-orientalis* |
| *G. aff. conica* |
| *G. atlantica* |
| *G. bulloides* |
| *G. falsostuarti* |
| *G. linneiana* |
| *G. mariei* |
| *G. stuartiformis* |
| *G. tricarinata* |
| *G. ventricosa* |
| *Globotruncana* sp. |
| *Globotruncanita* sp. |
| *Gta. elevata* |
| *Gta. Subspinosa* |
| *Radotruncana calcarata* |
| *r/K*-intermediates: |
| *Archaeoglobigerina* spp. |
| *A. cretacea* |
| *Archaeoglobigerina blowi* |
| *Globotruncanella havanensis* |
| *Ga. pschadae/* sp. |
| *"Globigerinelloides" bolli* |
| *"G." prairiehillensis* |
| *"G." ultramicrus* |
| *"Globigerinelloides"* sp. |
| *Pseudotextularia nutalli* |
| *r*-selected: |
| *Guembelitria* sp. |
| *Gublerina rajagopalani* |
| *Spiroplecta navarroensis* |
| *Planoheterohelix globulosa* |
| *Ph. punctulata* |
| *Ph. striata* |
| *Heterohelix* spp. |
| *Muricohedbergella holmdelensis* |
| *M monmouthensis* |
| *Laeviheterohelix* spp. |
| *Rugoglobigerina* sp. |

*Contusotruncana* and *Radotruncana*. The group *Globotruncana arca-lapparenti-orientalis* comprises several double keeled, biconvex taxa, and was, as expected, most frequently detected within the globotruncanid lineage. Other globotruncanid taxa, such as *G. linneiana, G. ventricosa. G. mariae, G. falsostuarti*, as well as *C. patelliformis* and *C. fornicata* are present throughout the section. The zonal marker—*Radotruncana calcarata*—is a comparatively rare element at Postalm that accounts for only a maximum of 1% of the assemblage.

The genus "*Globigerinelloides*" is present in numbers up to 15%, including "*G*". *bolli*, "*G*". *ultramicrus* and "*G*". *multispinus*. *Archaeoglobigerina* is mainly represented by two taxa, *A. cretacea* and *A. blowi*. Rugoglobigerinids (presumably *R. rugosa*?) are less abundant.

The planktic/benthic foraminifera ratio is very high throughout the section. Benthic foraminifera never display a higher share than 6%. Quantitative data show a peak in benthic foraminifera abundance in the lower part of the section (samples 7/38*K* and 7/39). High abundance in 7/39 is inferred by high numbers of tubular agglutinating taxa (presumably *Nothia* spp.).

### Life strategies of planktonic foraminifera

Upon examination of the >63 µm fraction, the foraminiferal assemblage displays opportunistic *r*-strategists as the dominant element, as small biserial and trochospiral taxa account for an overwhelming majority of individuals. K-strategists, which are exclusively represented by globotruncanids at Postalm, are mostly recorded with less than ten percent. Taxa showing a life strategy that cannot be clearly assigned, such as "*r/K*-intermediate" selected taxa such as "*Globigerinelloides*", show a similar frequency pattern as *K*-selected species. Table 2 shows the taxa recorded at the Postalm section and their life strategy. Figure 6 displays the distribution of taxa with respect to their life strategy and the inferred environmental characteristics.

## DISCUSSION

Examining foraminiferal assemblages from a bathyal environment of an active continental margin preserved in a mountain belt has some drawbacks. This study has to deal with poor preservation of microfossils due to a strong diagenetic overprint and minor folding and faulting of the sections. Still, records with restricted taxonomical resolution (especially with smaller foraminifera, i.e., ~63–125 µm) can give some indication of the palaeoecology and biostratigraphy; conspicuous biostratigraphic marker species are still clearly identifiable. If the loss of taxonomic information only permits the identification at the genus level, especially for small (<125 µm) morphotypes, comparison of the relative abundance of foraminiferal taxa is still possible. Likewise, the distribution of *r*- and *K*-strategists is a measurement that can typically be determined at genus level (*Hart, 1999*; *Premoli Silva & Sliter, 1999*; *Petrizzo, 2002a*; *Petrizzo, 2002b*; *Gebhardt et al., 2010*).

### Implications for biostratigraphy and palaeoenvironment

The *Radotruncana calcarata* Zone was first introduced by *Herm (1962)* and defines the interval between the first occurrence (FO) and the last occurrence (LO) of the nominate taxon.

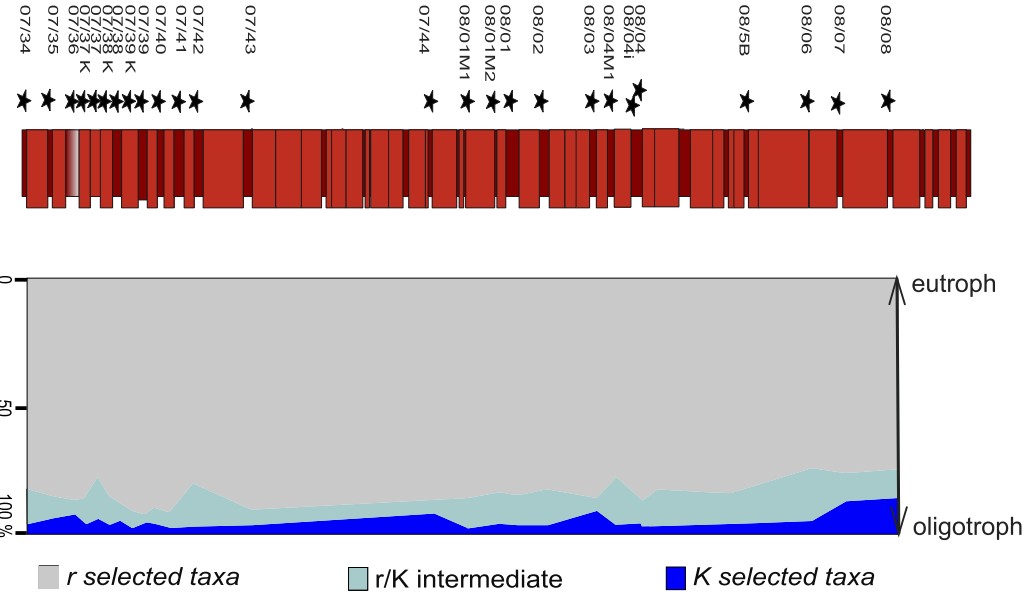

**Figure 6** **Frequency of foraminifera in respect to their ecological characteristics.** The vast majority of individuals in the >63 μm fraction assign to opportunistic *r*-selected taxa (grey), *r/K* intermediates (light blue) and *K*-selected taxa (dark blue) are represented by 10% each.

For a long time the top of the *R. calcarata* interval was used to define the Campanian—Maastrichtian boundary in plankton biostratigraphic zonations for the Late Cretaceous (e.g., *Salaj & Samuel, 1966*; *Caron, 1985*; *Sliter, 1989*). Today, chronostratigraphic correlations position this interval in the mid Campanian (*Robaszynski & Caron, 1995*— as *Globotruncana calcarata* TRZ, *Premoli Silva & Sliter, 1999*; *Berggren & Pearson, 2005*; *Huber, MacLeod & Tur, 2008*; *Robaszynski & Mzoughi, 2010*; *Ogg & Hinnov, 2012*).

Other studies concerned with the *Radotruncana calcarata* TRZ also recorded *G. angulata* (as *G.* cf. *angulata* in *Hart, 1987*), *Pseudoguembelitria costulata* (*Li & Keller, 1998*), *Rugoglobigerina hexacamerata*, "*Globigerinelloides*" *yaucoensis* (*Arz & Molina, 2001*), *Globotruncana aegyptiaca* (*Arz & Molina, 2001*; *Chacón, Martín-Chivelet & Gräfe, 2004*), "*Globigerinelloides*" *messinae* and *Pseudoguembelina costualta* (*Premoli Silva, Emeis & Robertson, 2005*) and *Globotruncana rosetta* (*Robaszynski & Mzoughi, 2010*). These taxa were not identified in the *R. calcarata* Zone at the Postalm section.

In our section from the Northern Calcareous Alps we seem to document the extinction of *Globotruncanita elevata* within the *R. calcarata* TRZ (see *Wolfgring, Hohenegger & Wagreich, in press*). Generally, the LO of this taxon is considered to have occurred shortly before or within the *R. calcarata* interval (e.g., *Robaszynski & Caron, 1995*; *Chacón, Martín-Chivelet & Gräfe, 2004*; *Cetean et al., 2011*; *Petrizzo, Falzoni & Premoli Silva, 2011*). *Wolfgring, Hohenegger & Wagreich (in press)* record this taxon to be a rare faunal element in the *R. calcarata* interval. *Globotruncanita elevata* does not appear in the quantitative analysis. Therefore, we assume a gradual disappearance of this taxon towards the middle Campanian in the Northern Calcareous Alps (and presumably all of the Tethyan realm). The zonal marker itself seems to displays an abrupt disappearance.

It is difficult to compare the results of different quantitative studies on Late Cretaceous planktonic foraminiferal assemblages with other locations, as different environments are studied (that display differences in the preservation of microfossils), and different methods are applied (starting with the examination of different size fractions and different ways of sample preparation). For instance, *Li & Keller (1998)* document an analysis of the foraminiferal assemblage in the >63 μm fraction from the South Atlantic DSDP site 525A (Walvis Ridge), together with an examination of the >105 μm fraction of Site 21 (Rio Grande Rise). The works of Petrizzo (2001), on planktonic foraminifera from Kerguelen Plateau, ODP Leg 183 and *Petrizzo (2002a)* and *Petrizzo (2002b)*, from Exmouth Plateau (ODP Sites 762 and 763), examine the >40 μm size fractions. Both the studies of *Li & Keller (1998)*, *Petrizzo (2002a)* and *Petrizzo (2002b)* discuss fully pelagic sections. *Arz & Molina (2001)* describe the foraminiferal fauna from the Tercis GSSP—this study examines the >106 μm size fraction from a shelf environment. *Elamri & Zaghbib-Turki (2014)* deal with the >100 μm fraction from a pelagic section recording the Santonian-Campanian boundary (Kalaat Senan area in Tunisia).

Data from the Postalm section show smaller planktonic foraminifera as the dominant element of the foraminiferal assemblage as the >63 μm was examined: Hedbergellids and small biserial planktonic foraminifera represent the vast majority in this size fraction. The proportion of benthic foraminifera and "larger" planktonic foraminifera, i.e., specialist taxa is very low in this size fraction in a hemipelagic to pelagic environment (*Yilmaz, 2008*; *Wagreich, Hohenegger & Neuhuber, 2012*; *Wolfgring, Hohenegger & Wagreich, in press*).

Foraminiferal assemblages in the Cretaceous Period are characterised with respect to the distinct sequence or succession of dominant planktonic foraminiferal taxa and lineages. During the Early Cretaceous, hedbergellids and towards the end of the Cretaceous Period, heterohelicids represented the dominant element in the planktonic foraminiferal communities of the open ocean's waters (*Hart, 1999*; *Premoli Silva & Sliter, 1999*). At Postalm, hedbergellids and heterohelicids, in varying numbers, still represent the vast majority of the foraminiferal assemblage.

We find a similar distribution of genera in other quantitative and semi-quantitative studies on Late Cretaceous communities. *Arz & Molina (2001)* correlate the *Rugoglobigerina hexacamerata* Zone at Tercis to the *R. calcarata* TRZ. Reflecting the distribution pattern, visible in the relative abundance of foraminiferal genera, the similarity to Postalm section is conspicuous, although palaeoenvironmental conditions are quite different, contrasting a pelagic bathyal setting to the Tercis shelf setting. There, heterohelicids and hedbergellids, together with globigerinelloidids are dominant elements whereas globotruncanids are represented by 10–20%. Rugoglobigerinids, which are almost absent at the Postalm section (only three samples yield rugoglobigerinids), are constantly present. Their proportion at Tercis increases towards the Maastrichtian (with almost 10%). We speculate that differences between pelagic and distal shelf planktonic foraminiferal communities in the mid- to Late Cretaceous are minimal. Postalm and Tercis (*Arz & Molina, 2001*) present both heterohelicids and hedbergellids as dominant faunal elements. Most individuals assigning to hedbergellids or heterohelicids are represented in smaller size fractions (<125 μm). The comparatively small share of these taxa recorded in *Arz & Molina (2001)* is presumably

due to the use of the 106 μm fraction for micropoalaeontological analyses. Therefore, a composition of the planktonic foraminiferal assemblage at Tercis similar to what is known from the Postalm section is very likely. The comparison of data from benthic and planktonic foraminifera from two different locations (a shelf environment at Tercis, and a bathyal slope at Postalm) shows that the composition of the planktonic foraminiferal community alone is not indicative of the palaeoenvironment. *Arz & Molina (2001)* report a high share of benthic foraminifera for the Tercis section (between 50 and 80%), while we record a maximum of six percent of benthic foraminifera at the Postalm section.

*Li & Keller (1998)* also report predominance of hedbergellids (*M. monmouthensis* and *M. holmdelensis*) around the *R. calcarata* interval at DSDP Site 21 (South Atlantic). Heterohelicid taxa are also represented in high numbers (*Ph. globulosa, Ph. planata, Laeviheterohelix pulchra* and *P. costulata*), and globotruncanids are represented there by 20%. As hedbergellids and heterohelicids predominate smaller size fractions (<125 μm), their comparatively high proportion of globotruncanids in the studies of *Li & Keller (1998)* and *Arz & Molina (2001)* might result from the use of the >105 μm fraction. The only rugoglobigerinid taxon present at Site 21, *Rugoglobigerina rugosa*, shows a discontinuous record during the Campanian at Site 21. The faunal composition recorded in *Li & Keller (1998)* gives information on the palaeogeographical and environmental setting. The palaeolatitude of DSDP sites 21 and 525A (36°S) and the lower to upper bathyal palaeodepth (*Moore et al., 1984*; *Li & Keller, 1998*) resemble the environmental setting we encounter at the Postalm section and so does the composition of the planktonic foraminiferal assemblage. The comparison of the foraminiferal assemblages at Postalm to *Arz & Molina (2001)* and *Li & Keller, (1998)* suggests that the palaeolatidudinal setting is more likely to influence the composition of the planktonic foraminiferal assemblages in open oceans in the Late Cretaceous than bathymetry.

The quantitative studies of Petrizzo (2001), *Petrizzo (2002a)* and *Petrizzo (2002b)*, both localities from the southern high latitudes, show few similarities in the relative abundance of taxa. Comparatively frew hedbergellid specimens were recorded at Site 183. The Upper Cretaceous assemblages of Site 183 display a very strong dominance of heterohelicids. The species *Ph. globulosa* alone sometimes accounts for 40% of the assemblage (Petrizzo, 2001), a feature that is not so prominently expressed in the foraminifera assemblages of Postalm section. Planktonic foraminiferal assemblages from the southern high latitudes, or the Austral margin are in many ways different from tropical, or mid latitude assemblages: a palaeoenvironment affected by cooler water masses is not only reflected in a special biostratigraphic scheme, but also in different dominant lineages (*Wonders, 1992*; *Huber, 1990*; *Petrizzo, 2002a*; *Petrizzo, 2002b*). A further comparison of the distribution pattern of foraminiferal lineages observed in quantitative studies from the southern high latitudes shows that hedbergellids are not as abundant, and globotruncanids are less diverse.

The semi-quantitative study of *Premoli Silva, Emeis & Robertson (2005)* also indicates similar abundance patterns as the Postalm section. While *Muricohedbergella holmdelensis* and *M. monmouthensis* are distributed equally at the Postalm section, the study from ODP Hole 160-967E only records *M. holmdelensis* as a common element during the *R. calcarata* interval.

## Benthic foraminifera

Benthic foraminifera appear only as rare faunal elements in quantitative data from the Postalm section but play a significant role for the reconstruction of the palaeoenvironment using presence–absence data (see *Wolfgring, Hohenegger & Wagreich, in press*). The Postalm section yields a highly diverse "Deep Water Agglutinating Foraminifera"—assemblage (*Kuhnt & Kaminski, 1990*), as well as abundant calcareous benthic foraminifera. Agglutinated genera like *Dorothia* or *Marssonella* occur together with abundant calcareous benthic foraminifera, especially nodosarids and lenticulinids. We interpret these assemblages based on the quantitative data presented here as typical Slope-marl assemblage or an upper to middle bathyal assemblage (following *Kuhnt, Kaminski & Moullade, 1989*; *Koutsoukos & Hart, 1990*; *Widmark & Speijer, 1997*; *Kaminski & Gradstein, 2005*).

Two minor peaks in benthic foraminifera abundance were recorded. These peaks are based on the high frequency of the taxon *Nothia* sp. However, as a result of this taxon's epifaunal mode of life (*Kuhnt, Kaminski & Moullade, 1989*; *Kuhnt & Kaminski, 1990*), mostly fragmented individuals were recovered and counted (which can lead to inaccurate results). Thus, we cannot eliminate the possibility that an accumulation of fragmented individuals of *Nothia* sp. was caused by episodic current activity rather than by a bloom in this taxon. However, if this taxon had indeed episodic blooms at the bottom of the bathyal slope basin reconstructed for the Postalm section, an increased flux of nutrients downslope would have positive influence on epifaunal detritivore species, such as *Nothia* (*Geroch & Kaminski, 1992*; *Kaminski & Gradstein, 2005*). These favourable palaeoecological conditions could have been triggered by several factors, e.g., turbiditic events, changes in bottom water currents, etc.

## Depositional water depths

The tectonic evolution of the Penninic oceanic realm as recorded by the Gosau Group sediments suggest certain constraints for the reconstruction of palaeodepths in parts of the Nierental Formation (*Wagreich & Krenmayr, 2005*; *Wagreich et al., 2009*).

The base of the Postalm section, as well as some other Gosau-sections record the transition from a neritic setting to a pelagic environment (*Wagreich & Krenmayr, 2005*; *Butt, 1981*). Changes in faunal composition reflect changes in the palaeoenvironment. To sketch a possible palaeodepth model several approaches were considered.

The methods of *Van der Zwaan, Jorissen & De Stigter (1990)* and *Hohenegger (2005)* both were applied. *Van der Zwaan, Jorissen & De Stigter (1990)* focus on quantitative data and the ratio of planktic and benthic foraminifera, while *Hohenegger (2005)* uses the possible depth ranges and presence–absence data of benthic foraminifera to calculate a possible palaeo waterdepth. The presence–absence data of bentic foraminifera used to calculate palaeo waterdepths using the method of *Hohenegger (2005)* was discussed in *Wolfgring, Hohenegger & Wagreich (in press)*.

The application of a planktic/benthic foraminifera ratio (*P/B*-ratio) is a popular—though sometimes unreliable method (see *Gibson, 1989*)—to estimate palaeo-water depths in modern, oligotrophic environments (*Van der Zwaan, Jorissen & De Stigter, 1990*; *Van der Zwaan et al., 1999*; *Gebhardt, Zorn & Roetzel, 2009*). With respect to the benthic

foraminiferal fauna at the Postalm section, we assume slightly dysoxic habitat conditions for benthic foraminifera (see *Wolfgring, Hohenegger & Wagreich, in press*). On that score, a mesotrophic regime should be taken into consideration (according to the TROX model by *Jorissen, De Stigter & Widmark, 1995*). Therefore, calculating palaeo waterdepths using the $P/B$-ratio without considering local environmental properties is likely to lead to inaccurate conclusions in this section (as, according to *Van der Zwaan et al., 1999*, $P/B$ ratios are sensitive to oxygen deficiency).

Results from the quantitative assessment show that a maximum of 6% of foraminifera recovered assign to benthic foraminiferal taxa. Thus, the application of the formula of *Van der Zwaan, Jorissen & De Stigter (1990)* would result in palaeo waterdepths around 1,200 m. This method has certain constraints—*Van der Zwaan, Jorissen & De Stigter (1990)* state that it is useful to estimate palaeodepths between 30 and 1,250 m. At Postalm section we record up to 100% planktonic foraminifera in standard quantitative data and therefore stretch this method to the limits as the application of a $P/B$ ratio in these samples is no longer possible using standard quantitative data.

The characteristics of the benthic foraminiferal communities resemble those of "Slope-Marl" assemblages (*Kuhnt, Kaminski & Moullade, 1989*; *Kaminski & Gradstein, 2005*), or "Upper to Middle Bathyal" communities (*Widmark & Speijer, 1997*). *Widmark & Speijer (1997)* document this particular assemblage type from various localities recording palaeo waterdepths from upper slope to abyssal depths. Applying the calculation method proposed in *Hohenegger (2005)* a mean (theoretical) depositional water depth of 695 m can be calculated. An average minimum water depth of 349 m at sample POST 7/35 and an average maximum water depth of 914 m at sample POST 6/07 were recorded. Although this method in its application to fossil and extinct taxa has severe limitations, and depth ranges for the Penninic Ocean active margin assemblages may differ considerably from estimates from the North Atlantic passive margin slope model, the estimates are within the principally inferred depth range. Figure 7 compares the two methods used for the calculation of palaeo-waterdepths at Postalm. The depth ranges of benthic foraminiferal taxa and the calculated palaeo-waterdepths for each sample can be found in Appendix S1.

In addition to the information provided by the benthic foraminiferal record, valuable data are also provided on the assessment of the composition of the planktic foraminiferal assemblage:according to data from planktonic foraminifera we consider the Penninic Ocean during the Campanian-Maastrichtian a non-restricted environment in terms of faunal exchange. The assemblage recorded at Postalm neither seems to lack essential elements of a planktonic foraminiferal community, nor can we record any hints towards an endemic foraminiferal fauna in the Penninic Ocean.

Rugoglobigerinids are a rare faunal element at the Postalm section. Apart from the preference for warmer water layers (as suggested by *Abramovich et al., 2003*; *Falzoni et al., 2014*; *Petrizzo et al., 2015*), *Olsson (1977)*, *Hart (1980)* and *Georgescu (2005)* speculate on *Rugoglobigerina* as a taxon preferring shallow water as rugoglobigerinids are frequently a common element or even dominant in planktonic foraminifera assemblages in shallow water deposits (e.g., epeiric seas, neritic environments). Thus, this fact also supports the reconstruction of a hemipelagic to pelagic setting in the *R. calcarata* Zone. The Postalm

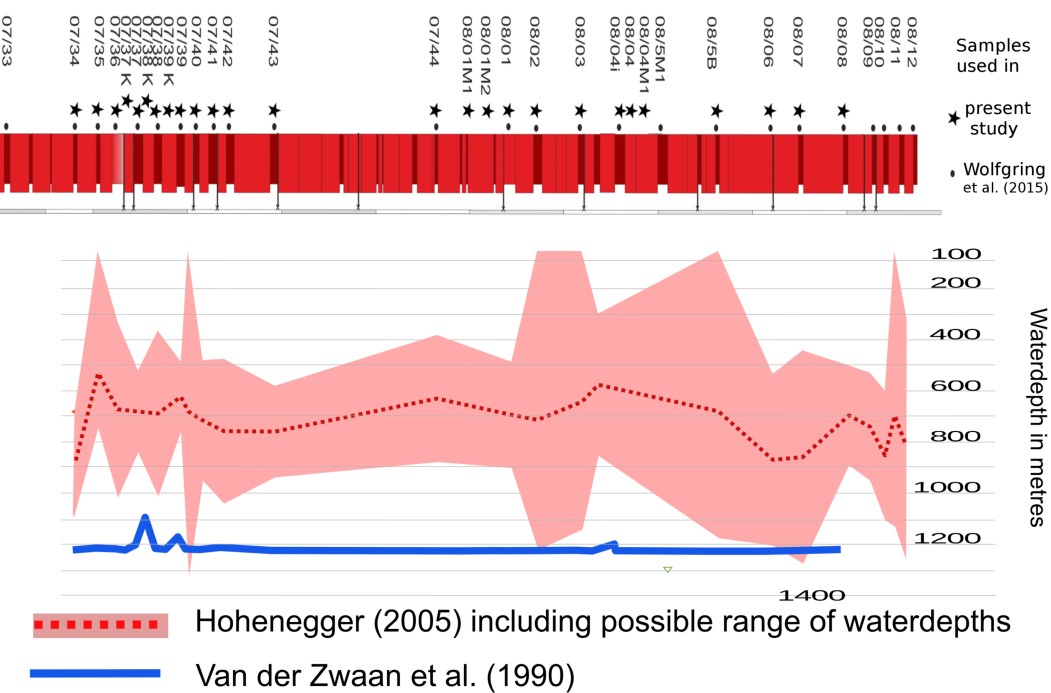

**Figure 7** **An average palaeo-waterdepth was calculated applying the methods of** *Hohenegger (2005)* **and** *Van der Zwaan, Jorissen & de Stigter (1990)*. The benthic foraminiferal presence-absence dataset assessed in *Wolfgring, Hohenegger & Wagreich (in press)* was applied to calculate palaeo-waterdepths after the method of *Hohenegger (2005)*. Quantitative data assessed in this study was used to calculate the waterdepth using *Van der Zwaan, Jorissen & de Stigter (1990)*.

section displays a sparse record of the genus *Rugoglobigerina*. On examination of the isotopic signatures of this taxon, a habitat in the upper, warmer layers of the ocean appears likely (*Abramovich et al., 2003*; *Falzoni et al., 2014*; *Petrizzo et al., 2015*).

Summarising the information on foraminiferal assemblages and on the tectonic evolution of the active margin of the Penninic ocean (see *Butt, 1981*; *Wagreich, 1993*), we can reconstruct the depositional environment during the *R. calcarata* Zone as an upper to middle slope setting with palaeo-waterdepths of at least 500–800 m. We interpret the minor differences in calculations of water depths within a depth range of 300 m using the approach by *Hohenegger (2005)* as artificial, being a result of the low (and thus sometimes erratic) numbers of benthic foraminifera recorded and the inaccuracy of depth habitat estimates for Cretaceous foraminifera. Thus, both applied quantitative methods of palaeodepth estimates are not able to record and resolve 3rd order sea-level changes which may be in the range of up to 75 m within the mid Campanian (*Haq, 2014*).

## Palaeoecology

Cretaceous ocean systems are characterised by well stratified water masses, offering niches for a variety of life strategies (*Leckie, 1989*; *Huber & Watkins, 1992*; *Price et al., 1998*; *Norris et al., 2001*; *Leckie, Bralower & Cashman, 2002*). Reconstructing the palaeoecology of planktonic foraminifera from pelagic environments mostly relies on the use of recent analogues in morphotypes (as found in *Hemleben, Spindler & Anderson, 1989*). Indications

of the environmental properties, prevailing in the preferred habitat of planktonic foraminifera, can be found by the examination of stable isotopes (e.g., *D'Hondt & Arthur, 1995*; *Price et al., 1998*; *Abramovich et al., 2003*). The variety of life strategies is neither fully understood yet, nor easy to summarise. Nontheless, *Premoli Silva & Sliter (1999)* apply the ecological concept of *K*- and *r*-strategists for Cretaceous planktonic foraminifera.

*K*-strategists represent specialist taxa that thrive in oligotrophic enviroments. This group is often represented by keeled forms assigned to the *Globotruncana* and *Globotruncanita* lineage. The ecological characteristics of K-strategists and the comparison of the functional morphology of keels to recent analogues suggest an interpretation of globotruncanids as deep-dwelling forms, favouring colder waters and requiring an oligotrophic environment (*Hart, 1980*; *Premoli Silva & Sliter, 1999*). Nevertheless, there are a number of examples of keeled forms that lived in the mixed layer at shallower water depths; thus, there is no generally accepted interpretation of the functional morphology of keels in Cretaceous planktonic foraminifera (*Huber, 1990*; *Huber, Hodell & Hamilton, 1995*; *D'Hondt & Arthur, 1995*; *Abramovich et al., 2003*).

Heterohelicids are considered opportunistic taxa (*r*-strategists), indicating unstable conditions and generally preferring eutrophic environments, and are presumed to be indicators for stressful environments (*Leckie, 1987*; *Nederbragt, 1991*; *Premoli Silva & Sliter, 1999*; *BouDagher-Fadel, 2013*). It is speculated that this group thrives in the oxygen minimum zone—a model that explains the interpretation of heterohelicid dominance within an assemblage, sees this group as indicating a locally well-developed oxygen minimum zone (*Leckie et al., 1998*; *Pardo & Keller, 2008*; *Reolid et al., 2015*).

By examining the habitat patterns of planktonic species during the latest Cretaceous (mid Campanian to late Maastrichtian), the study of *Abramovich et al. (2003)* interprets some heterohelicids as inhabitants of the subsurface layers, or water masses close to the thermocline (*Planoheterohelix globulosa, Ph. planata, Ph. punctulata*).

Hedbergellids are as well interpreted as *r*-strategists and are generally considered open marine species (*Leckie, 1987*; *Koutsoukos & Hart, 1990*; *Norris & Wilson, 1998*; *Premoli Silva & Sliter, 1999*; *Petrizzo, 2002a*; *Petrizzo, 2002b*; *Gebhardt, 2004*; *Bornemann & Norris, 2007*), and exhibit similar ecological preferences as heterohelicids. Studies on planktonic foraminifera that integrated information from stable isotope data, interpret hedbergellids as surface dwellers, occupying the upper mixed layer (e.g., *Price et al., 1998*; *Fassell & Bralower, 1999*; *Norris et al., 2002*; *Ando et al., 2009*; *Ando, Huber & MacLeod, 2010*). *Norris & Wilson (1998)*, *Petrizzo (2002a)* and *Petrizzo (2002b)* suggested a wider depth distribution for mid-Cretaceous hedbergellids. *Ando, Huber & MacLeod (2010)* present evidence that *H. delrioensis* migrated from a shallow to a deep mixed habitat during the mid-Cenomanian. *Huber et al. (2011)* indicate hedbergellids to be highly flexible and to show a dynamic behaviour. *Gebhardt et al. (2010)* characterise hedbergellids from the Cenomanian to Turonian of the Austrian Alps as intermediate forms between *r* and *K* strategists. As with most biserial planktonic foraminifera, the trochospiral hedbergellids have been considered as opportunistic taxa that prefer eutrophic environments and occupy the upper mixed layer (*Premoli Silva & Sliter, 1999*; *Gebhardt, 2004*).

The genus *Schackoina* is often considered as an indicator for poorly oxygenated environments, but its life strategy has not been sufficiently investigated (*Magniez-Jannin, 1998*; *Premoli Silva & Sliter, 1999*; *Petrizzo, 2002a*; *Petrizzo, 2002b*). Therefore, we exclude this taxon from palaeoecological analyses.

With the ongoing evolution of more complex morphotypes as a driving force, the relative abundance of *K*- and *r*-strategists follows a distinct pattern throughout the Cretaceous (*Hart, 1980*; *Leckie, 1989*; *Premoli Silva & Sliter, 1999*). According to *Premoli Silva & Sliter (1999)*, Late Cretaceous planktonic foraminiferal communities are, in contrast to foraminiferal communities from the Early Cretaceous, highly diversified and dominated by *K*-selected taxa.

The distribution of *r*- and *K*-selected taxa does not only provide information on the stratigraphical age and palaeoecological regime. The relationship between *r*- and *K*-selected taxa can be characteristic for the latitudinal distribution. With a palaeolatitude of approximately 35° N, we consider Postalm, and the north-western Tethys in general, to represent a low to mid-latitude setting.

Quantitative data helped to document at least 15 planktonic foraminiferal genera. Therefore, we consider the assemblage at Postalm as diverse and dominated by *r*-selected taxa. The number of *K*-selected specialists diminishes polewards and *r*-selected taxa prevail (*Premoli Silva & Sliter, 1999*) but no similarities to a species- or morphogroup-distribution pattern known from higher latitudes, boreal assemblages (i.e., chalk facies), were identified. Postalm section yields single and double keeled *K*-selected taxa that are typical elements of Campanian tropical to mid-latitude foraminiferal communities. At the Postalm section, *K*-selected specialists, which are dominant in low latitude faunas, are present, but only few in number. Therefore, the foraminiferal assemblage at Postalm, which includes Tethyan taxa, is typical of the Transitional Realm *s* mid-latitudes (following *Sliter, 1977*; *Malmgren, 1991*; *Huber, 1992*; *Premoli Silva & Sliter, 1999*; *Nishi et al., 2003*). Compared to low-latitude assemblages from Tunisia or Italy, Postalm section displays fewer *K*-strategists (i.e., globotruncanids) and more opportunistic taxa (hedbergellids, heterohelicids).

## Implications for palaeoceanography of the mid campanian Penninic Ocean

Some works describe the Penninic Ocean (or the Alpine Tethys) as a restricted environment during the mid-Cretaceous (e.g., *Mort et al., 2007*; *Gebhardt et al., 2010* from the Cenomanian/Turonian). The Late Cretaceous foraminiferal assemblage examined in this study shows all fundamental elements of a well-developed low to mid-latitude planktonic foraminiferal community. Although a few taxa recorded in other studies were not identified at Postalm, the planktonic foraminiferal assemblage with its high diversity (*Wolfgring, Hohenegger & Wagreich, in press*) does not give indications for a restricted oceanic environment. From the investigated section we record members of the *Globotruncana* lineage as alleged deep dwellers, heterohelicids, which are reported to thrive in oxygen minimum zones and in upper surface waters, as well as the Hedbergellidae that mostly preferred surface water habitats. In light of the Cretaceous sea-level maximum at the

Cenomanian-Turonian boundary (see *Haq, 2014*) we interpret the increased ventilation of the Penninic Ocean as a result of tectonic processes that opened seaways from the southern main Tethys Ocean system into the northwestern Tethys and its continuation into the Atlantic, probably due to plate tectonic rearrangements and subsidence events from the Turonian to Campanian (e.g., *Wagreich, 1993*; *Reicherter & Pletsch, 2000*).

Considering the low to mid-latitude setting of the Northern Calcareous Alps, the frequency and distribution of taxa and ecological groups approaches results from other studies on Late Cretaceous planktonic foraminiferal assemblages from bathyal or hemipelagic to pelagic sections (e.g., *Chacón, Martín-Chivelet & Gräfe, 2004*; *Robaszynski & Mzoughi, 2010*; *Elamri & Zaghbib-Turki, 2014*). Furthermore, the cyclostratigraphically dated synchronous appearance and disappearance of the zonal marker fossil *R. calcarata* in the Alpine sections and in Tunisia indicates good connections to the tropical Tethys Ocean (*Robaszynski & Mzoughi, 2010*; *Wagreich, Hohenegger & Neuhuber, 2012*).

Foraminiferal data from Postalm give little information on fluctuations in sea-level during the *R. calcarata* interval. There are some minor changes, easily overlooked in standard quantitative data, and/or hard to interpret at the fringes of the assemblages in the 1–3 percentage range. Changes in the relative abundance of keeled globotruncanids towards the top of the section and the continuous presence of *Schackoina* in the stratigraphically younger part of the *R. calcarata* interval could indicate subtle changes in the palaeoceanography of the Penninic Ocean than a robust sea-level or water-depth signal.

A similar situation is recorded from other proxy data from this section: minor carbon isotope peaks (*Wagreich, Hohenegger & Neuhuber, 2012*; *Wendler, 2013*) or geochemical proxy data (*Neuhuber et al., 2015*), could also imply small scale changes in sea-level but have to be interpreted with caution (*Neuhuber et al., 2015*).

We conclude that although major changes and cycles (sequences) in the range of several Ma may influence foraminiferal communities, those short-term changes within the 800 ka long *R. calcarata* Zone had minimal impact on planktonic foraminiferal communities in a well connected, bathyal setting of water depths over 500 m.

## Differences in limestones and marls and the preservation of microfossils—how does cyclic sedimentation affect the foraminiferal record?

The effects of diagenesis on the cyclic pelagic rock record is a widely discussed subject (e.g., *Westphal & Munnecke, 2003*). Postalm shows limestone—marly limestone alternations that were interpreted to reflect precession cycles (*Wagreich, Hohenegger & Neuhuber, 2012*). For this study, both stronger indurated marly limestones as well as marls from within the same precession cycle were disintegrated with hydrogen peroxide and tensides, and subsequently examined quantitatively for foraminifera. In contrast to marly samples, firm foraminifera packstone required repeated cooking in hydrogen peroxide, as well as the application of tensides to dissolve. Marly samples were dissolved following the standard preparation procedures.

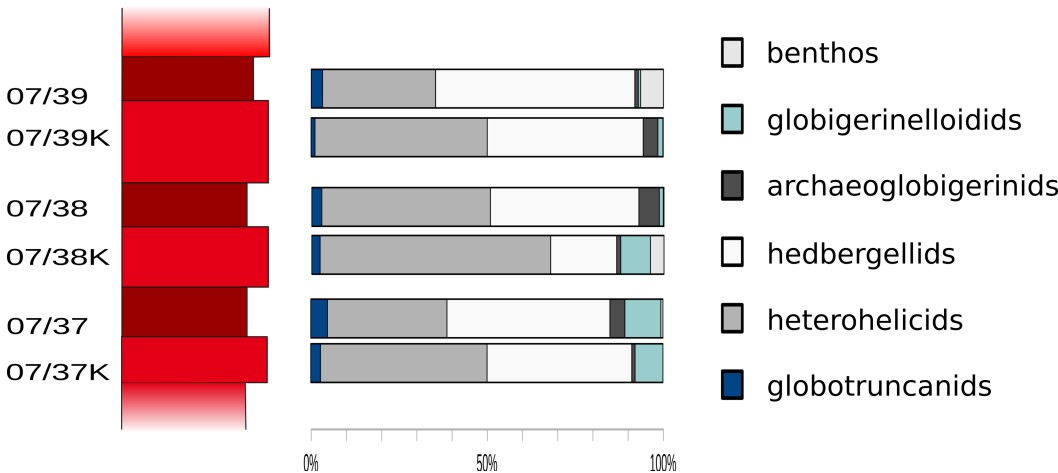

**Figure 8 Foraminiferal assemblages in limestone/marl couplets.** Frequencies of globotruncanids, heterohelicids, hedbergellids, archaeoglobigerinids, globigerinelloidids and benthic foraminifera from limestone-marl couplets. Biserial planktonic taxa are more abundant in the firmer limestone samples.

Figure 8 displays differences in the composition of foraminiferal assemblages between samples from firm marly limestone and samples from softer marls. No trends pointing towards a significant diagenetic influence on the foraminiferal community are recorded in those parts of a sedimentary cycle.

A slight shift towards compact biserial microfossils could be significant in one sample (07/38). In general, heterohelicids are more frequently recorded in higher numbers in samples from firmer carbonates than in samples from marls—in the three limestone-marl couplets examined here, the numbers of biserial planktonic taxa exceed the average abundance of heterohelicids present in all other samples of this section, as well as the average number of heterohelicids calculated for these three couplets.

However, most abundance data recorded from limestones give results within the standard deviations calculated for each taxonomic group (i.e., globotruncanids, globigerinelloidids, heterohelicids, hedbergellids and benthic foraminifera) in marls. No difference in species diversity was observed, which also argues against a significant diagenetic impact on foraminiferal assemblages. Furthermore, no evident signs of dissolution in either taxonomic group in foraminifera can be found throughout the section that displays a palaeoenvironment that is well above the CCD.

Nevertheless, the fact that rhythmic limestone-marl alternations are likely to represent an orbital influence on climate should not be overlooked (e.g., *Bernet, Eberli & Anselmetti, 1998*; *Sageman et al., 1998*; *Westphal, Böhm & Bornholdt, 2004*). Thus, significant changes in the abundance and frequency of groups of microfossils do not necessarily need to be explained by diagenesis but may reflect changing environmental conditions during orbital cycles as does the changing lithology. Precession cycles result from changes in insolation, which have a considerable ecological impact. In this study we compare few examples from the "margins" of orbital cycles and believe that subtle changes in foraminiferal assemblages could also be influenced by changes in the ecological conditions. For instance, could the fact

that all samples from firm carbonates record higher heterohelicid abundance have resulted from changes in the extent of the oxygen minimum zone at the end (or the beginning) of a precession cycle due to changes in detrital input and plankton productivity (similar patterns were observed in planktonic foraminifera assemblages in sapropel-cycles from the Mediterranean: e.g., *Sierro et al., 1997*).

Moreover, results from comparison of abundance data from marls and marly limestones suggest that dissolution effects on microfossils were either the same or absent in both lithologies.

## CONCLUSIONS

The evaluation of planktonic foraminiferal communities (>63 μm) from the mid Campanian *R. calcarata* Taxon Range Zone, recorded in rhythmic limestone—marl alternations, at Postalm section (Northern Calcareous Alps, Austria) gives detailed information on the behaviour of planktonic communities within a well-defined time frame.

Although microfossils exhibit a moderate to poor state of preservation, the main characteristics of foraminiferal communities could be tracked. In particular, for constraining the age and biozonation of the sequence, the prominent zonal marker fossil, *R. calcarata* is considered a reliable marker in Late Cretaceous biostratigraphy, despite its rare occurrence.

Morphotypes and ecological groups in planktonic foraminifera were recorded, permitting speculations on the depositional environment and palaeoecology. The Postalm section displays a foraminiferal assemblage that is characteristic of hemipelagic to pelagic sequences, with dominance of $r$-selected opportunistic taxa, predominantly represented by heterohelicids and hedbergellids. $K$-selected specialist taxa represent approximately 10% of the assemblage. The same applies to "$r/K$ intermediate" taxa (such as globigerinelloidids). The planktonic foraminiferal community from Postalm displays a typical mid-latitude distribution of taxonomic groups.

Minor fluctuations in the distributional pattern of foraminiferal genera have been recorded. However, no distinct trends or significant events and no significant difference between the general assemblage structure in marls and marly limestones could be recognized. Therefore, diagenesis had a minor influence, and lithological cycles are interpreted as having been triggered mainly by insolation-induced climate cycles.

All major foraminiferal taxonomic groups and a broad spectrum of ecological strategies were recognised from the Late Cretaceous foraminiferal assemblages at Postalm. Therefore, we assume an unrestricted environment for the Campanian Penninic Ocean, with open connections to the Tethyan seaway.

## ACKNOWLEDGEMENTS

Reviews by an anonymous reviewer and Brian Huber greatly improved the quality of the manuscript.

### Funding

Funding provided by the Austrian Science Fund FWF, project P24044-N24 Campanian Orbital Cyclostratigraphy [CampOC], UNESCO IGCP 609 Climate-environmental deteriorations during greenhouse phases: Causes and consequences of short-term Cretaceous sea-level changes. The funders had no role in study design, data collection and analysis, decision to publish, or preparation of the manuscript.

### Grant Disclosures

The following grant information was disclosed by the authors:
Austrian Science Fund FWF: P24044-N24.
Campanian Orbital Cyclostratigraphy [CampOC]: UNESCO IGCP 609.

### Competing Interests

The authors declare there are no competing interests.

### Author Contributions

- Erik Wolfgring conceived and designed the experiments, performed the experiments, analyzed the data, wrote the paper, prepared figures and/or tables.
- Michael Wagreich contributed reagents/materials/analysis tools, reviewed drafts of the paper.

### Data Availability

The raw data generated for this study are counts of microfossils and are included in the manuscript.

### Supplemental Information

Supplemental information for this article can be found online at http://dx.doi.org/10.7717/peerj.1757#supplemental-information.

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
