# Peer review of "A quantitative look on northwestern Tethyan foraminiferal assemblages, Campanian Nierental Formation, Austria"

_PeerJ, doi:10.7717/peerj.1757_

## Round 0.1 · original submission · Major Revisions

· Academic Editor

Major Revisions

I apologise for the long review period, it was challenging to find reviewers, but now we have two broadly positive reviews, but both suggest changes. A third potential reviewer would not provide a full review until additional acknowledgement of the recent literature was included. ' I noticed that the taxonomy of planktonic foraminifera is out of date. Authors apparently overlooked quite few papers published since 2009 in well known journals like Journal of Foraminiferal Research.'
Please address this omission in the referencing, as well as the two additional reviews for PeerJ to consider publication.

·

Basic reporting

.

Experimental design

.

Validity of the findings

.

Additional comments

The paper is generally well written and provides a useful, detailed study across a well resolved stratigraphic interval and new insight to the paleoceanography of the northern calcareous alps region. The authors need to pay attention to use of appropriate genus names (e.g., Macroglobigerinelloides is not valid as the name was never published in the peer-reviewed literature as far as I know; Hedbergella is now restricted to taxa that became extinct at the end of the Aptian; Heterohelix is now restricted to an odd monospecific form (americana) that has subtrianangular to subrectangular chambers). Also, the calcarata zone is in the mid-Campanian, not late Campanian as the title and text state. Generally the interpretation of the population changes is fine, though I think the use of r- vs. K-selected taxa and assumption that heterohelicids are indicators of oxygen minimum conditions are an oversimplification that have become embedded in the literature.

Reviewer 2 ·

Basic reporting

Figure captions need some development.

Experimental design

The research question needs to be more clearly set-up in the introduction placing this work in the wider context and stating why it was necessary - see comments attached.

Further information on methods required re. disaggregation of sediments etc.

Validity of the findings

I'd like the raw count data to be provided in Table 1 rather than the authors calculated percentages to maximize transparency and the versatility of the dataset.

The new abundance data need to be more efficiently discussed and utilized in the interpretations.

Include discussion of how your findings of the palaeonvironment compare with what was already known about the site, e..g, paleowater depth inferences of 500 - 800 m in W&W, 2015. Is it just support for interpretation from an independent source? be specific.

Additional comments

This manuscript investigates the composition of foraminiferal communities across a short-lived biozone (R. calcarata) in the late Cretaceous to better understand the palaeoenvironmental setting of the site and background variability in assemblages outside of events.

Define high-resolution (~X kyrs) because this means different things to different people

Relevance of third sentence in abstract?

Carefully read whole document for English

Why not diagenetic control on assemblage compositions?

Do you really mean palaeocology here? All the ecology stuff seems to be drawn from references elsewhere – it’s rather that you’re using the ecology to infer the environment isn’t it?

Two shorts in the same sentence at end of abstract

What palaeoceanographic and climatic changes?

L30 – when is the late campanian? Help the reader out with some numbers/figure etc.

L30 – careful using ‘period’ in this context. The Campanian is not a period. Better to use ‘interval’ or such instead

L51 – Ref. figure 1 here

The introduction needs to be much better set-up to make it clear why this study was necessary and what question you’re trying to address. Understanding the variability of background conditions is clearly important but you need to say why.

L64 – Need to put this interval of study in the context of the wider Cretaceous timescale and climate for the reader. Yes, the Campanian is a time of change but is your time interval? What is happening climatically within this interval? is there known to be much variability? Why did you pick it specifically?

Should mention the findings of Wolfgring et al (2015) paper here in the introduction in more detail as an intro to your study rather than a tag-on at the end of the sentence. For instance, I can’t obviously see any reference to the fact that there is remarkably little evolutionary turnover in planktic foram assemblages in the same section/interval beyond the zonal marker itself. Make it clear why it was so important to look at abundance of species when the diversity was already known from the earlier paper. This should also be integrated with the discussion on your remarkably static assemblages in terms of species distributions as well as diversity.

L116 – a few more methodological details here please. For instance, how long do you ‘cook the samples for? This will be helpful to other workers trying to extract foraminifera from equally well-lithified materials.

For clarity please indicate in the text the number of additional samples here that expanded the earlier works.

Please be specific that both planktic and benthic foraminifera were counted in each sample.

L136 – why were two different water depth estimates made? Can you please mention briefly in the main text pros/cons of each approach to justify?

L145 – which taxa did you explicitly exclude? Can you refer to a table or list the taxa?

L156 – Some SEM’s or light microscope images would help here. What does moderate to poor mean? The scale has shifted with the discovery of glassy forams. Can you please specify – recrysallised? Etc.

Results – so how many species/genera did you identify in your samples?

L221 - “at the Postalm section”

L240 – note to biostrat here but refer to as presented in W&W, 2015

Biostrat. section - the discussion about R. elongata occurring with R. calcarata zone featured in W&W, 2015. You need to discuss what your new data tells you about this event. For instance can you say anything about extinction patterns to add to this information etc.

L270 – Yes it is difficult when people apply different methods but surely some statement can be made. How does the planktic foram diversity compare with other sites? Because of the size fraction/preservation at various sites is your site higher, lower…

L274 – new paragraph not necessary here as still talking about size and different workers.

L271 -284 - This whole section on different workers using different size fractions is very listy and not really that important to the story as you’ve already said that it’s difficult to make comparisons because of different environments and size fractions studies. I’d also add preservation to this list.

L289 – do they dominate globally? Low latitudes? Open ocean?

L291 – repeat of information from opening sentence – delete

L308 – So based on the similarities in composition between taxa what can you conclude?

L314 – what does this mean for the palaeonvrinoemnt presumably the southern high lats looked very different even in the Cretaceous to the tropics?

L324 – Be clear why you are communicating this information by providing context. For instance you don’t provide the ecology of the other more dominant taxa above just this one rare species. Why? Why do we care about the Rugoglobigerinids?

L329 – are quantitative counts of benthics in the Cretaceous generally rare or just in your dataset. Please clarify as sentence meaning unclear. What significant role?

L333 – are the identified genera agglutinated? If so, please say so.

L343 – why does their epifaunal mode of life make them less likely to be preserved? How big did a fragment have to be, to be counted? Is it possible that you used less than half the original specimen and thus, doubled the count?

L370 – unclear sentence meaning

L376 – because above 1250 you’d expect only planktic foraminifera?

L396 – remove paragraph and just refer reader to caption and figure in paragraph above.

L408 – what do these workers mean by shallow? This is where more context for this group earlier would help to frame if say the open ocean sections were rarer in this species compared to shallow shelf sections.

L412 – why have you discounted the estimates based on the Van de Zwaan estimate? There were problems with both techniques.

This section is titled biostratigraphy and implications. As far as I can tell it is a comparison between sites. The implications of the findings are not realized in this section at all.

L429 – nope in fact morphology is often a terrible way to infer environment think of the contrasting ecology of the keeled disc forms in the Eocene versus today. Remove reference to ‘always’.

L437 – ‘stressful’?

L450 – ‘assigned’

L452 – is this assignment based purely on analogy to modern? If so, consider the Eocene Morozovella – these are keeled surface dwellers and there are no keeled deep forms at this time. Why are the Cretaceous forms more analogous to the modern than say the Eocene ones?

Ok – information about the ecology of each group is being given out in dribs and drabs here which is difficult for the reader to follow particularly given the constant switching between the genera being discussed, e..g L448 Hedbergellids, next para on Globotruncana and then back to Hedbergellids, and the repetition of information from different authors.

L459 - Already mentioned surface habitat of Hedbergellids in 447 from same source

L472 – ok so what is the palaeoecological regime at this site? State it clearly based on the evidence.

L475 – specifically with what kind of latitudinal pattern – can you please be explicit so we can see if assemblage matches the interpretation.

L485 – ok I can see some of this info right at the end now – keep it together where you discuss do K or R stategists dominate and latitudinal stuff in the latitudinal paragraph

L509 – “all-Cretaceous”?

L515 – said difficulties earlier ditch this sentence

L533 – relevance of this paragraph to the discussion? If you keep make it clear why this information is being provided.

L537 – it’s probably quite important to be clear that this is within the resolution of the data, which involves both preservational and temporal biases that may result in aliasing of the signal. For instance, if precession say dominates the climatic signal at this site then you wouldn’t expect to see variability within your section.

L555 – state no diagenetic impact then in next line say what the differences are. Present the differences first then draw a conclusion.

A more general setup to the climate of the time is necessary earlier on

L558 - Could it be preferential dissolution cycles?

L592 -594 – statement on R. calcarata is odd here include in previous paragraph or not at all.

Appendix 1 – does ‘1’ refer explicitly to the presence rather than no. of individuals.

L1129 – can you please be very clear which data are from the earlier paper and the new datapoints here

Where are all the depth ranges from?

Can you add the depths into the presence/absence table as an extra column on the end to keep all the info together?

Figure 1 – Caption is incomplete. Need to mention inset. Note that site location is shown by large star. What’s the small star near the top right of the Salzburg label. Missing the pink/yellow units in key.

Figure 2 – Again refer to the grey star in the caption to help the reader. Define the acronym N.C.A. on key/caption

Figure 3 – italicize fossil name. Again the caption is insufficient. It should tell the reader the main reason for the figure. Indicate that the black stars show sample locations.

Figure 4 – “overview of”

Figure 5 – change in section now red and dark red unlike Fig. 4. Be consistent

Benthic foraminifera should be different colour to Hedbergella to avoid confusion.

I found this figure really difficult to follow the abundance axes even if you don’t include numbers can you please at least include tick marks.

“Benthic foraminifera” shouldn’t be italicized.

Figure 6 – 10% of each?
Why was the same dataset not used in both reconstructions?
Make sure figure labels are rotated on axes for easy reading

Figure 8 – blue shading doesn’t match in the key (dark blue) and figure (bright blue)

---

## Round 0.2 · accepted · Accept

· Academic Editor

Accept

You have addressed the reviewers comments so that the paper is now ready for publication.